# IMPUS: Image Morphing with Perceptually-Uniform Sampling Using Diffusion Models

**Zhaoyuan Yang**[1*], **Zhengyang Yu**[2*], **Zhiwei Xu**[2], **Jaskirat Singh**[2], **Jing Zhang**[2],
**Dylan Campbell**[2], **Peter Tu**[1], **Richard Hartley**[2]
GE Research[1]     Australian National University[2]
{zhaoyuan.yang,tu}@ge.com   {zhengyang.yu,zhiwei.xu,jaskirat.
singh,jing.zhang,dylan.campbell,richard.hartley}@anu.edu.au

## Abstract

We present a diffusion-based image morphing approach with perceptually-uniform sampling (IMPUS) that produces smooth, direct and realistic interpolations given an image pair. The embeddings of two images may lie on distinct conditioned distributions of a latent diffusion model, especially when they have significant semantic difference. To bridge this gap, we interpolate in the locally linear and continuous text embedding space and Gaussian latent space. We first optimize the endpoint text embeddings and then map the images to the latent space using a probability flow ODE. Unlike existing work that takes an indirect morphing path, we show that the model adaptation yields a direct path and suppresses ghosting artifacts in the interpolated images. To achieve this, we propose a heuristic bottleneck constraint based on a novel relative perceptual path diversity score that automatically controls the bottleneck size and balances the diversity along the path with its directness. We also propose a perceptually-uniform sampling technique that enables visually smooth changes between the interpolated images. Extensive experiments validate that our IMPUS can achieve smooth, direct, and realistic image morphing and is adaptable to several other generative tasks.

## 1 Introduction

In various image generation tasks, diffusion probabilistic models (DPMs) (Ho et al., 2020; Song et al., 2021a;b) have achieved state-of-the-art performance with respect to the diversity and fidelity of the generated images. With certain conditioning mechanisms (Dhariwal & Nichol, 2021; Ho & Salimans, 2022; Rombach et al., 2022), DPMs can smoothly integrate guidance signals from different modalities (Zhang & Agrawala, 2023; Yang et al., 2023) to perform conditional generation. Among different controllable signals, text guidance (Rombach et al., 2022; Saharia et al., 2022; Nichol et al., 2022) is widely used due to its intuitive and flexible user control.

Different from other generative models, such as generative adversarial networks (GANs) (Goodfellow et al., 2014), DPMs have some useful properties such as invertibility. Firstly, ODE samplers (Lu et al., 2022a;b; Liu et al., 2022) can approximate a bijective mapping between a given image $\mathbf{x}_0$ and its Gaussian latent state $\mathbf{x}_T$ by connecting them with a probability flow ODE (Song et al., 2021b). Secondly, the conditioning signal can be inverted with textual inversion (Gal et al., 2023) which can be used to obtain a semantic embedding of an image. These properties make DPMs useful for image manipulation, including the tasks of image editing (Meng et al., 2022; Brooks et al., 2023; Hertz et al., 2023), image-to-image translation (Tumanyan et al., 2023; Parmar et al., 2023) and image blending (Lunarring, 2022; Avrahami et al., 2023; Pérez et al., 2023).

In this paper, we consider the task of image morphing, generating a sequence of smooth, direct, and realistic interpolated images given two endpoint images. Morphing (Dowd, 2022; Nri, 2022; Zope & Zope, 2017), also known as image metamorphosis, changes object attributes, such as shape or texture, through seamless transitions from one frame to another. Image morphing is usually

---

*Equal contribution, order determined via coin flip and may be listed in either order. Code is available at:
https://github.com/GoL2022/IMPUS

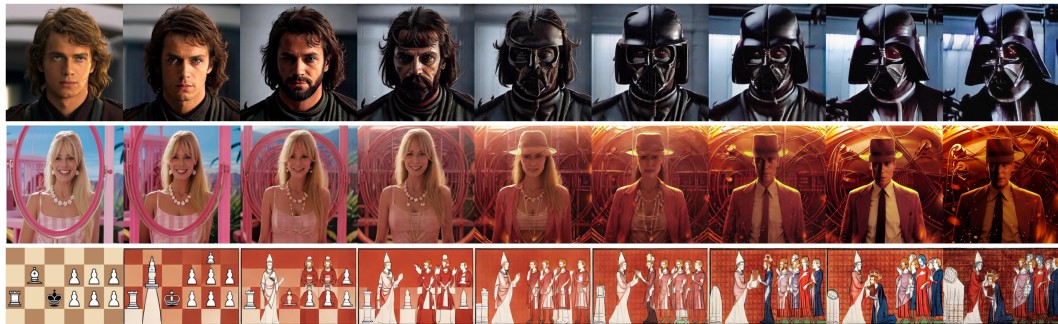

Figure 1: Image Morphing with IMPUS. IMPUS achieves smooth, real, and direct interpolation between two images with perceptually-uniform transition. More results in Sec. 5 and Appendix.

achieved by using image warping and color interpolation (Wolberg, 1998; Lee et al., 1996), where the former applies a geometric transformation with feature alignment as a constraint, and the latter blends image colors. This has been well-studied in computer graphics (Lee et al., 1996; Seitz & Dyer, 1996; Zhu et al., 2007; Wolberg, 1998; Liao et al., 2014; Rajković, 2023; Zope & Zope, 2017; Shechtman et al., 2010) but less so in computer vision where the starting point is natural images.

Inspired by the local linearity of the CLIP (Radford et al., 2021) text embedding space (Liang et al., 2022; Kawar et al., 2023) and the probability flow ODE formulation of DPMs (Song et al., 2021a), we propose the first image morphing approach with perceptually-uniform sampling using diffusion models (IMPUS). A similar setting was explored by Wang & Golland (2023), which shows strong dependency on pose guidance and fine-grained text prompts, while our approach achieves better performance on smoothness, directness, and realism, without additional explicit knowledge, such as human pose information, being required. Some examples are visualized in Figure 1.

We summarize our main contributions as: (1) we propose a robust open world automatic image morphing framework with diffusion models; (2) we discover that low-rank adaptation contributes significantly to morphing task; (3) we present an algorithm for perceptually-uniform sampling that encourages the interpolated images to have visually smooth changes; (4) we propose three evaluation criteria, namely smoothness, realism, and directness, with associated metrics to measure the quality of the morphing sequence; and (5) we explore potential applications of morphing in broader domains such as video interpolation, model evaluation as well as data augmentation.

## 2 RELATED WORK

**Text-to-image Generation.** The mainstream of current text-to-image generation models (Nichol et al., 2022; Rombach et al., 2022; Saharia et al., 2022) are based on diffusion models (Ho et al., 2020; Song et al., 2021a;b), especially the conditional latent diffusion models (Rombach et al., 2022). Given the flexibility of textual prompting (Radford et al., 2021), text-to-image generation is well studied to generate both high-quality and open world images (Ramesh et al., 2022). Text inversion Gal et al. (2023) enables capturing specific concepts present in real-world images.

**Image Editing.** Among image editing models (Meng et al., 2022; Kawar et al., 2023; Brooks et al., 2023; Hertz et al., 2023), text-guided image editing is a burgeoning field due to the advances in language models (Brown et al., 2020; Chowdhery et al., 2023; Touvron et al., 2023). Editing for real images is enabled by latent state inversion (Song et al., 2021a; Mokady et al., 2023). Unlike image editing that requires specific guidance on local or global factor of variation. Image morphing has no requirements for guidance on particular subject or attribute variations between two different images.

**Image-to-image Translation.** Image-to-image translation (Tumanyan et al., 2023; Parmar et al., 2023; Su et al., 2023; Kim et al., 2023; Cheng et al., 2023) is usually done to transfer an image from source domain to a target domain while preserving the content information. Unlike image-to-image translation, which seeks to find a mapping between two domains, image morphing aims to achieve seamless and photo-realistic transitions between them.

**Image Morphing.** Human involvement is usually required for conventional image morphing (Lee et al., 1996; Seitz & Dyer, 1996; Zhu et al., 2007; Wolberg, 1998; Liao et al., 2014; Dowd, 2022). Nri (2022) uses neural networks to learn the morph map between two images, which however fails to obtain a smooth morphing of images that have a large appearance difference. Alternatively, Miller & Younes (2001); Rajković (2023); Shechtman et al. (2010); Fish et al. (2020) achieve image morphing by minimizing the path energy on a Riemannian manifold to obtain geodesic paths between the initial and target images, and perceptual as well as similarity constraints across the neighboring frames are enforced to achieve seamless image transitions. Most existing morphing works either require human intervention or are restricted to a narrow domain (see Appendix B.2 for details). In this work, we present a novel diffusion morphing framework that achieves image morphing with minimum human intervention and is capable of applying to arbitrary real-world images.

## 3 BACKGROUND

### 3.1 PRELIMINARIES OF DIFFUSION MODELS

**Denoising Diffusion Implicit Models (DDIMs).** With some predefined forward process $\{q_t\}_{t\in[0,T]}$ that gradually adds noises to an image $\mathbf{x}_0$, a diffusion model approximates a corresponding reverse process $\{p_t\}_{t\in[0,T]}$ to recover the image from the noises. In this work, we use a DDIM (Song et al., 2021a), see Appendix A for more details. Given $q(\mathbf{x}_t \mid \mathbf{x}_0) := \mathcal{N}\left(\mathbf{x}_t; \sqrt{\beta_t}\mathbf{x}_0, (1-\beta_t)\mathbf{I}\right)$ and a parameterized noise estimator $\epsilon_\theta$, we have the following update rule in the reverse diffusion process

$$\mathbf{x}_{t-1} = \sqrt{\beta_{t-1}}\underbrace{\left(\frac{\mathbf{x}_t - \sqrt{1-\beta_t}\epsilon_\theta^{(t)}(\mathbf{x}_t)}{\sqrt{\beta_t}}\right)}_{\text{``predicted } \mathbf{x}_0\text{''}} + \underbrace{\sqrt{1-\beta_{t-1}-\sigma_t^2}\epsilon_\theta^{(t)}(\mathbf{x}_t)}_{\text{``direction pointing to } \mathbf{x}_t\text{''}} + \underbrace{\sigma_t\epsilon_t}_{\text{``random noise''}} , \quad (1)$$

where $\sigma_t$ is a free variable that controls the stochasticity in the reverse process. By setting $\sigma_t$ to 0, we obtain the DDIM update rule, which can be interpreted as the discretization of a continuous-time *probability flow ODE* (Song et al., 2021a; Lu et al., 2022a). Once the model is trained, this ODE update rule can be reversed to give a deterministic mapping between $\mathbf{x}_0$ and its latent state $\mathbf{x}_T$ (Dhariwal & Nichol, 2021), given by

$$\frac{\mathbf{x}_{t+1}}{\sqrt{\beta_{t+1}}} - \frac{\mathbf{x}_t}{\sqrt{\beta_t}} = \left(\sqrt{\frac{1-\beta_{t+1}}{\beta_{t+1}}} - \sqrt{\frac{1-\beta_t}{\beta_t}}\right)\epsilon_\theta^{(t)}(\mathbf{x}_t). \quad (2)$$

A DDIM can be trained using the same denoising objective as DDPM (Ho et al., 2020). With the sampled noise $\epsilon$ and a condition vector $\mathbf{e}$, it is given by

$$\min_\theta \mathcal{L}_{\text{DPM}}(\mathbf{x}_0^{(0)}, \mathbf{e}; \theta) = \min_\theta \mathbb{E}_{\mathbf{x}_t \sim q(\mathbf{x}_t|\mathbf{x}_0), t\sim\mathcal{U}(1,T), \epsilon\sim\mathcal{N}(\mathbf{0},\mathbf{I})}\left[\|\epsilon_\theta(\mathbf{x}_t, t, \mathbf{e}) - \epsilon\|_2^2\right]. \quad (3)$$

**Classifier-free Guidance (CFG).** When a conditional and an unconditional diffusion model are jointly trained, at inference time, samples can be obtained using CFG (Ho & Salimans, 2022). The predicted noises from the conditional and unconditional estimates are defined as

$$\hat{\epsilon}_\theta(\mathbf{x}_t, t, \mathbf{e}) := w\epsilon_\theta(\mathbf{x}_t, t, \mathbf{e}) + (1-w)\epsilon_\theta(\mathbf{x}_t, t, \varnothing), \quad (4)$$

where $w$ is the guidance scale that controls the trade-off between mode coverage as well as sample fidelity and $\varnothing$ is a null token used for unconditional prediction.

### 3.2 DESIDERATA FOR IMAGE MORPHING

The image morphing problem can be formulated as a constrained Wasserstein barycenter problem (Simon & Aberdam, 2020; Agueh & Carlier, 2011; Chewi et al., 2020), where two images are interpreted as two distributions $\xi^{(0)}$ and $\xi^{(1)}$ respectively. The optimal solution to this problem is an $\alpha$-parameterized family of distributions given by

$$\xi^{(\alpha)} = \arg\min_\nu (1-\alpha)\mathcal{W}_2^2\left(\xi^{(0)}, \nu\right) + \alpha\mathcal{W}_2^2\left(\xi^{(1)}, \nu\right), \quad (5)$$

where $\mathcal{W}_2(\cdot, \cdot)$ is the 2-Wasserstein distance (Villani, 2016), $\alpha \in [0, 1]$ is the interpolation parameter, $\mathcal{M}$ is the target data manifold and $\nu$ is a probability measure on $\mathcal{M}$. The parameterized distribution $\xi^{(\alpha)}$ connects $\xi^{(0)}$ and $\xi^{(1)}$ (Villani, 2021), which leads to a smooth and direct transition process. The formulation is informative but it is challenging to generalize such formulation to real world high dimensional images; thus, we explore an alternative, tractable method, without losing track of the desirable characteristics from such formulation. Based on the objective of image morphing in existing works, we propose three important properties for high-quality image morphing.

**Smoothness.** Transition between two consecutive images from the generated morphing sequence should be seamless. Different from traditional image morphing methods that consider a physical transition process (Arad et al., 1994; Lee et al., 1995) to achieve seamless transitions, we aim to estimate a *perceptually* smooth path with a less stringent constraint.

**Realism.** The interpolated images should be visually realistic and distributed on a target manifold for image generation models. In other words, the images should be located in high-density regions according to the diffusion model.

**Directness.** Similar to optimal transport that finds the most efficient transition between two probability distributions, the morphing transition should be direct and have as little variation along the path as possible under the constraints of smoothness and realism. A desirable and direct morphing process should well fit human intuition for a minimal image morph.

## 4 Image Morphing with Perceptually-uniform Sampling

Given two image latent vectors $\mathbf{x}_0^{(0)}$ and $\mathbf{x}_0^{(1)}$, corresponding to an image pair, we aim to generate the interpolated sequence $\{\mathbf{x}_0^{(\alpha)}\}$ with $\alpha \in [0, 1]$ such that it can well satisfy the three criteria in Sec. 3.2. In the following, we denote an image latent encoding by $\mathbf{x}_0 \in \mathbb{R}^n$, a conditional text embedding by $\mathbf{e} \in \mathbb{R}^m$, and a diffusion timestep by $t \in \mathbb{Z}$. We define a diffusion model parameterized by $\theta$ as $\epsilon_\theta : \mathbb{R}^n \times \mathbb{Z} \times \mathbb{R}^m \to \mathbb{R}^n$, where the forward and reverse processes are defined in Sec. 3.1. In Sec. 4.1, we propose a semantically meaningful construction between the two images in the given image pair by interpolating their textual inversions $\{\mathbf{e}^{(0)}, \mathbf{e}^{(1)}\}$ and latent states $\{\mathbf{x}_T^{(0)}, \mathbf{x}_T^{(1)}\}$. Then, in Sec. 4.2, we show that model adaptation with a novel heuristic bottleneck constraint can control the trade-off between the directness of the morphing path and the quality of interpolated images. In Sec. 4.3, we introduce a perceptually-uniform sampling technique.

### 4.1 Interpolation in Optimized Text-Embedding and Latent-State Space

**Interpolating Text Embeddings.** Inverting the concepts in each image into optimized text embeddings with a pre-trained DPM facilitates modeling conditioned distributions that are centered around the input images. Inspired by recent works (Liang et al., 2022; Kawar et al., 2023) showing that the CLIP feature space is densely concentrated and locally linear, we choose to linearly interpolate between optimized CLIP text embeddings $\mathbf{e}^{(0)}$ and $\mathbf{e}^{(1)}$ of the image pair. To address potential nonlinearity & discontinuity between semantically distinct samples, given $\mathbf{x}_0^{(0)}$ and $\mathbf{x}_0^{(1)}$, we initialize the embedding $\mathbf{e}$ by encoding a simple common prompt, set as <*An image of* [*token*]>, where [*token*] is either the root class of the pair, e.g., "bird" for "flamingo" and "ostrich", or the concatenation of the pair, e.g., "beetle car" for "beetle" and "car". Since two embeddings are initialized by a common prompt, they stay closed after optimization, which is important for our locally linear assumption. The optimized text embeddings can be obtained by optimizing the DPM objective in Eq. (3) as

$$\mathbf{e}^{(0)} = \arg\min_{\mathbf{e}} \mathcal{L}_{\text{DPM}}(\mathbf{x}_0^{(0)}, \mathbf{e}; \theta) \quad \text{and} \quad \mathbf{e}^{(1)} = \arg\min_{\mathbf{e}} \mathcal{L}_{\text{DPM}}(\mathbf{x}_0^{(1)}, \mathbf{e}; \theta) . \tag{6}$$

The optimized text embeddings $\mathbf{e}^{(0)}$ and $\mathbf{e}^{(1)}$ can well represent the concepts in the image pair. Particularly, the conditional distributions $p_\theta(\mathbf{x}|\mathbf{e}^{(0)})$ and $p_\theta(\mathbf{x}|\mathbf{e}^{(1)})$ closely reflect the degree of image variation of the image pair, where the images are highly possible to be under their associated conditional distributions. Due to the local linearity and continuity of the CLIP embedding space, $p_\theta(\mathbf{x}|\mathbf{e}^{(0)})$ and $p_\theta(\mathbf{x}|\mathbf{e}^{(1)})$ can be smoothly bridged by linearly interpolating between the optimized embeddings, i.e., $p_\theta(\mathbf{x}|\mathbf{e}^{(\alpha)}) := p_\theta(\mathbf{x} \mid (1 - \alpha)\mathbf{e}^{(0)} + \alpha\mathbf{e}^{(1)})$, providing an alternative to the intractable problem of Eq. (5). See Appendix E.1 for analysis on effects of common prompt.

**Interpolating Latent States.** With the optimized text embeddings, we compute $p_\theta(\mathbf{x}|\mathbf{e}^{(\alpha)})$ to smoothly connect the two conditional distributions associated with the image pair. We then present latent state interpolation between the two image $\mathbf{x}_0^{(0)}$ and $\mathbf{x}_0^{(1)}$. We find that a vanilla interpolation between $\mathbf{x}_0^{(0)}$ and $\mathbf{x}_0^{(1)}$ leads to severe artifacts in the interpolated images. Hence, we interpolate in the latent distribution $\mathcal{N}(\mathbf{x}_T; 0, I)$ by first deterministically mapping the images to the distribution via the probability flow ODE in Sec. 3.1. As shown in Khrulkov et al. (2023), the diffusion process between $\mathbf{x}_0$ and $\mathbf{x}_T$ can be seen as an optimal transport mapping, which suggests that latent state similarity can also reflect certain pixel-level similarity (see Appendix E.2 for details), making it essential to correctly interpolate the latent states. Following previous work (Wang et al., 2023a), we smoothly interpolate between $\mathbf{x}_0^{(0)}$ and $\mathbf{x}_0^{(1)}$ via spherical linear interpolation (slerp) to their latent states $\mathbf{x}_T^{(0)}$ and $\mathbf{x}_T^{(1)}$ to obtain an interpolated latent state $\mathbf{x}_T^{(\alpha)} := \frac{\sin(1-\alpha)\omega}{\sin\omega}\mathbf{x}_T^{(0)} + \frac{\sin\alpha\omega}{\sin\omega}\mathbf{x}_T^{(1)}$, where $\omega = \arccos(\mathbf{x}_T^{(0)\mathsf{T}}\mathbf{x}_T^{(1)}/(\|\mathbf{x}_T^{(0)}\|\|\mathbf{x}_T^{(1)}\|))$. The denoised result $\mathbf{x}_0^{(\alpha)}$ is then retrieved via the probability flow ODE by using the interpolated conditional distribution $p_\theta(\mathbf{x}|\mathbf{e}^{(\alpha)})$.

## 4.2 MODEL ADAPTATION WITH A HEURISTIC BOTTLENECK CONSTRAINT

Interpolating in the optimized text embedding and latent state space using the forward ODE in Sec. 4.1 facilitates a faithful reconstruction of images in the given image pair and a baseline morphing result in Wang & Golland (2023). However, this approach leads to severe image artifacts when the given images are semantically different, as shown in Sec. 5.1. We also observe that this baseline approach often fails to generate direct transitions by transiting via non-intuitive interpolated states.

To obtain direct and intuitive interpolations between the images in the given image pair, we introduce model adaptations on the image pair. This can limit the degree of image variation in the morphing process by suppressing high-density regions that are not related to the given images. A standard approach is to fine-tune the model parameters using the DPM objective in Eq. (3) by $\min_\theta \mathcal{L}_{\text{DPM}}(\mathbf{x}_0^{(0)}, \mathbf{e}^{(0)}; \theta) + \mathcal{L}_{\text{DPM}}(\mathbf{x}_0^{(1)}, \mathbf{e}^{(1)}; \theta)$. However, this vanilla fine-tuning leads to object detail loss, as shown in Sec. 5.2, indicating that the conditional density of the source model is prone to collapse to the image modes without being aware of the variations of their image regions. To alleviate this issue, we introduce model adaptation with a heuristic bottleneck constraint. We use the low-rank adaptation (LoRA) (Hu et al., 2022) approach, which helps to retain mode coverage and reduces the risk of catastrophic forgetting (Wang et al., 2023b) due to its low-rank bottleneck. The optimization problem is given by

$$\min_{\Delta\theta_e} \mathcal{L}_{\text{DPM}}(\mathbf{x}_0^{(0)}, \mathbf{e}^{(0)}; \theta + \Delta\theta_e) + \mathcal{L}_{\text{DPM}}(\mathbf{x}_0^{(1)}, \mathbf{e}^{(1)}; \theta + \Delta\theta_e), \text{ s.t. rank}(\Delta\theta_e) = r_e, \quad (7)$$

where LoRA rank is set to be much smaller than the dimension of model weights. We find that the ideal rank w.r.t. quality of the morphs is based on a trade-off between *directness* and *fidelity*. Directness is affected by the *sample diversity* between endpoint images (e.g., low sample diversity occurs in image pairs that are intra-class or conceptually connected via a shared concept; large sample diversity occurs in image pairs that are inter-class or conceptually related via a root class such as "mammal" for "lion" and "camel"). *Fidelity* is affected by the *perceptual gap* between endpoints, i.e., a larger perceptual gap between the endpoints can lead to higher risk in quality loss after model adaptation with a larger rank due to mode collapse. Typically, a larger LoRA rank can enhance the directness of morphs by supressing sample diversity, but may lead to low-fidelity results (e.g., over-smooth morphs), which underscores the importance of finding a proper trade-off.

**Relative Perceptual Path Diversity.** Based on these qualitative observations, we define *relative perceptual path diversity* (rPPD) to evaluate the diversity along the morphing path, relative to the diversity of the endpoint images (see Appendix E.7 for rationale). To achieve this, we use our baseline morphing result from Sec. 4.1 and compute the average perceptual difference between the consecutive pairs, relative to the perceptual difference between the endpoints, given by

$$\gamma = \frac{\sum_{i=1}^{N-1} \text{LPIPS}\left(\mathbf{x}^{(\alpha_i)}, \mathbf{x}^{(\alpha_{i+1})}\right)}{(N-1)\,\text{LPIPS}\left(\mathbf{x}^{(0)}, \mathbf{x}^{(1)}\right)}, \quad (8)$$

where $\text{LPIPS}(\cdot, \cdot)$ measures the perceptual difference between two images (Zhang et al., 2018) and $\alpha_i = i/(N-1)$ for $i \in [0, N-1]$ and $N$ samples along the path. Empirically, we set the LoRA rank to $r_e = 2^{\max(0, \lfloor 18\gamma - 6 \rfloor)}$, where $\lfloor \cdot \rfloor$ returns the rounddown integer.

**Unconditional Bias Correction.** At inference time, to achieve high text alignment, a widely adopted strategy is to set a CFG scale $w > 1$ as in Eq. (4). While the aforementioned model adaptation strategy uses conditioned images and text embeddings, the unconditional distribution modeled by the same network is also changed. Due to parameter sharing, however, it causes an undesired bias. To address this, we consider two strategies: 1) randomly discard conditioning (Ho & Salimans, 2022) 2) separate LoRA parameters for fine-tuning the unconditional branch on $\mathbf{x}^{(0)}$ and $\mathbf{x}^{(1)}$. We observe that the latter can provide more robust bias correction. Hence, we use additional LoRA parameters $\Delta\theta_0$ with a small rank $r_0$, i.e.,

$$\min_{\Delta\theta_0} \mathcal{L}_{\text{DPM}}(\mathbf{x}_0^{(0)}, \varnothing; \theta + \Delta\theta_0) + \mathcal{L}_{\text{DPM}}(\mathbf{x}_0^{(1)}, \varnothing; \theta + \Delta\theta_0), \text{ s.t. rank}(\Delta\theta_0) = r_0 . \tag{9}$$

For inference, with $\theta_e = \theta + \Delta\theta_e$ and $\theta_0 = \theta + \Delta\theta_0$, we parameterize the noise prediction as

$$\hat\epsilon_\theta\left(\mathbf{x}_t, t, \mathbf{e}\right) := w\epsilon_{\theta_e}\left(\mathbf{x}_t, t, \mathbf{e}\right) + (1 - w)\epsilon_{\theta_0}\left(\mathbf{x}_t, t, \varnothing\right) . \tag{10}$$

### 4.3 PERCEPTUALLY-UNIFORM SAMPLING

In order to produce a smooth morphing sequence, it is desirable for consecutive samples to have similar perceptual differences. However, uniform sampling along the interpolated paths is not guaranteed to produce a constant rate of perceptual changes. We introduce a series of adaptive interpolation parameters $\{\alpha_i\}$. The core idea is to use binary search to approximate a sequence of interpolation parameters with constant LPIPS differences. Details are provided in Appendix D.1.

## 5 EXPERIMENTS

**Evaluation Metrics.** We validate the effectiveness of our proposed method with respect to the three desired image morphing properties defined in Sec. 3.2: 1) *Directness.* For a morph sequence with $N$ consecutive interpolation parameters $\alpha_i \in [0, 1]$, we report the total LPIPS, given by $\sum_{i=0}^{N} \text{LPIPS}(x^{(\alpha_i)}, x^{(\alpha_{i+1})})$, and the maximum LPIPS to the nearest endpoint, given by $\max_i \min(\text{LPIPS}(x^{(\alpha_i)}, x^{(0)}), \text{LPIPS}(x^{(\alpha_i)}, x^{(1)}))$, as indicators of directness. 2) *Realism.* We report FID (Heusel et al., 2017) to measure the realism of generated images. However, we note that this is an imperfect metric because the number of morph samples is limited and the samples are not i.i.d., leading to a bias. 3) *Smoothness.* We report the perceptual path length (PPL) (Karras et al., 2019) under uniform sampling, defined as $\text{PPL}_\varepsilon = \mathbb{E}_{\alpha \sim U(0,1)}\left[\frac{1}{\varepsilon^2} \text{LPIPS}(\mathbf{x}^{(\alpha)}, \mathbf{x}^{(\alpha+\varepsilon)})\right]$ where $\varepsilon$ is a small constant. For our perceptually-uniform sampling strategy, the total LPIPS measure is appropriate for evaluating smoothness (see Appendix F.1 for rationale). For a fair comparison with Wang & Golland (2023), we also use their evaluation setting: 1) the total LPIPS from 16 uniformly-spaced pairs ($\text{PPL}_{1/16}$) and 2) the FID between the training and the interpolated images.

**Datasets.** The data used comes from three main sources: 1) benchmark datasets for image generation, including *Faces* (50 pairs of random images for each subset of images from CelebA-HQ (Karras et al., 2018)), *Animals* (50 pairs of random images for each subset of images from AFHQ (Choi et al., 2020), including *dog*, *cat*, *dog-cat*, and *wild*), and *Outdoors* (50 pairs of *church* images from LSUN (Yu et al., 2015)), 2) internet images, e.g., the flower and beetle car examples, and 3) 25 image pairs from Wang & Golland (2023).

**Hyperparameters for Experiments.** We set the LoRA rank for unconditional score estimates to 2, and the default LoRA rank for conditional score estimates is set to be heuristic (auto). The conditional parts and unconditional parts are finetuned for 150 steps and 15 steps respectively. The fine-tune learning rate is set to $10^{-3}$. Hyperparameters for text inversion as well as guidance scales vary based on the dataset. See Appendix D for more details.

Table 1: Comparison with baseline.

|  | $\text{PPL}_{1/16} \downarrow$ | FID $\downarrow$ |
|---|---|---|
| Wang & Golland (2023) | 143.25 | 198.16 |
| IMPUS (ours) | 92.58 | 148.45 |

### 5.1 COMPARISON WITH EXISTING TECHNIQUES

For a fair comparison with Wang & Golland (2023), we use the same evaluation metrics and the same set of images and show the performance in Table 1. Our approach has a clear advantage w.r.t.

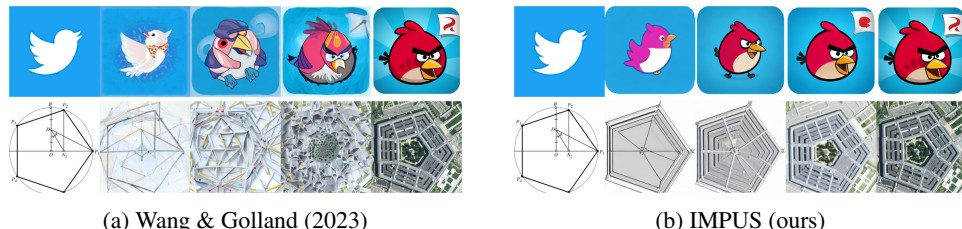

(a) Wang & Golland (2023)  (b) IMPUS (ours)

Figure 2: Compared with Wang & Golland (2023), our method achieves better sample consistency and quality. See Appendix for more trajectories to see comparison in smoothness.

Table 2: Quantitative results on benchmark. The lower the better for all metrics. For endpoint reconstruction errors, we report the mean and maximum (in parentheses) of LPIPS at the endpoints.

| | Text Inv. | Auto Rank | Rank 4 | Rank 32 | LPIPS$_T$ | LPIPS$_M$ | FID | PPL | Endpoint Error |
|---|---|---|---|---|---|---|---|---|---|
| | | ✓ | | | 1.13 | **0.37** | 48.89 | 248.06 | 0.092 (0.308) |
| female↔male | ✓ | ✓ | | | 1.16 | 0.39 | 48.20 | **238.87** | 0.081 (0.137) |
| | ✓ | | ✓ | | 1.14 | 0.38 | 46.84 | 322.31 | 0.079 (0.132) |
| | ✓ | | | ✓ | **1.09** | **0.37** | **46.12** | 367.70 | 0.080 (0.169) |
| | | ✓ | | | 1.61 | 0.50 | 46.28 | **451.85** | 0.125 (0.206) |
| cat↔dog | ✓ | ✓ | | | 1.75 | 0.52 | 45.82 | 485.48 | 0.119 (0.167) |
| | ✓ | | ✓ | | 1.71 | 0.52 | 48.11 | 613.77 | 0.119 (0.245) |
| | ✓ | | | ✓ | **1.60** | **0.49** | **43.21** | 804.36 | 0.116 (0.190) |
| | | ✓ | | | **1.46** | **0.39** | 35.71 | **533.70** | 0.113 (0.412) |
| church↔church | ✓ | ✓ | | | 1.79 | 0.42 | 36.52 | 1691.48 | 0.093 (0.248) |
| | ✓ | | ✓ | | 1.85 | 0.42 | 36.02 | 1577.45 | 0.088 (0.289) |
| | ✓ | | | ✓ | 1.66 | 0.41 | **34.15** | 2127.56 | 0.090 (0.266) |

both metrics. A qualitative comparison is shown in Figure 2, where our method has better image consistency and quality with severe artifacts as in Wang & Golland (2023) given two drastically different images (see first row of images). See Appendix B for a further comparison.

## 5.2 ADDITIONAL RESULTS AND ABLATION STUDIES

We perform a detailed analysis of the proposed approach with respect to different configurations and hyperparameters. Figure 3 shows the effectiveness of our method on CelebA-HQ, AFHQ, and LSUN-Church. Total LPIPS (LPIPS$_T$), Max LPIPS (LPIPS$_M$), and FID are calculated based on the sampling with a constant LPIPS interval of 0.2. PPL is with 50 intervals uniformly sampled in [0, 1].

**Ablation Study on Textual Inversion.** Given the importance of the text prompt for accurate conditional generation, we have claimed that suitable text embedding is critical for image morphing, which is optimized via text inversion. Alternatively, the optimized e could be replaced by a fine-grained human-designed text prompt.

For the benchmark datasets, we select the subset of female↔male, cat↔dog, and church↔church. We ablate textual inversion by using a prompt that describes the categories and show results in Table 2. For female↔male, we use the prompt "a photo of woman or man". For cat↔dog, we use the prompt "a photo of dog or cat". In Table 2, without textual inversion, morphing sequences have a shorter and more direct path (smaller LPIPS$_T$ and LPIPS$_M$) but have a larger reconstruction error at the endpoints. A poor reconstruction caused by the lack of textual inversion may lead to inferior morphing quality, see Appendix E.5 for details. Thus, text inversion with a coarse initial prompt is more robust.

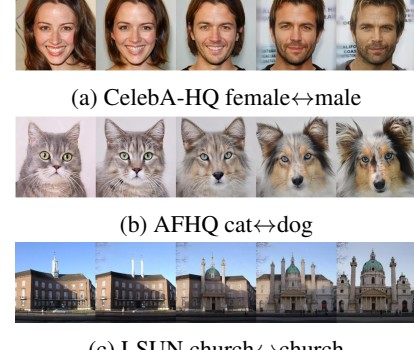

(a) CelebA-HQ female↔male

(b) AFHQ cat↔dog

(c) LSUN church↔church

Figure 3: Benchmark examples.

Table 3: Ablation study on model adaptation with text inversion. All metrics are the lower the better.

| | Adaptation | LPIPS$_T$ | LPIPS$_M$ | FID | PPL | Endpoint Error |
|---|---|---|---|---|---|---|
| CelebA-HQ | | $1.45 \pm 0.05$ | $0.41 \pm 0.01$ | $52.76 \pm 4.28$ | $196.30 \pm 41.78$ | $0.192 \pm 0.083$ |
| | ✓ | $1.07 \pm 0.07$ | $0.37 \pm 0.01$ | $43.59 \pm 5.16$ | $201.87 \pm 26.34$ | $0.083 \pm 0.001$ |
| AFHQ | | $2.27 \pm 0.20$ | $0.55 \pm 0.03$ | $42.67 \pm 12.39$ | $414.10 \pm 50.76$ | $0.254 \pm 0.007$ |
| | ✓ | $1.90 \pm 0.18$ | $0.54 \pm 0.03$ | $36.79 \pm 11.87$ | $679.96 \pm 159.04$ | $0.130 \pm 0.007$ |
| LSUN | | $1.94$ | $0.40$ | $42.48$ | $663.33$ | $0.184$ |
| | ✓ | $1.79$ | $0.42$ | $36.52$ | $1691.48$ | $0.093$ |

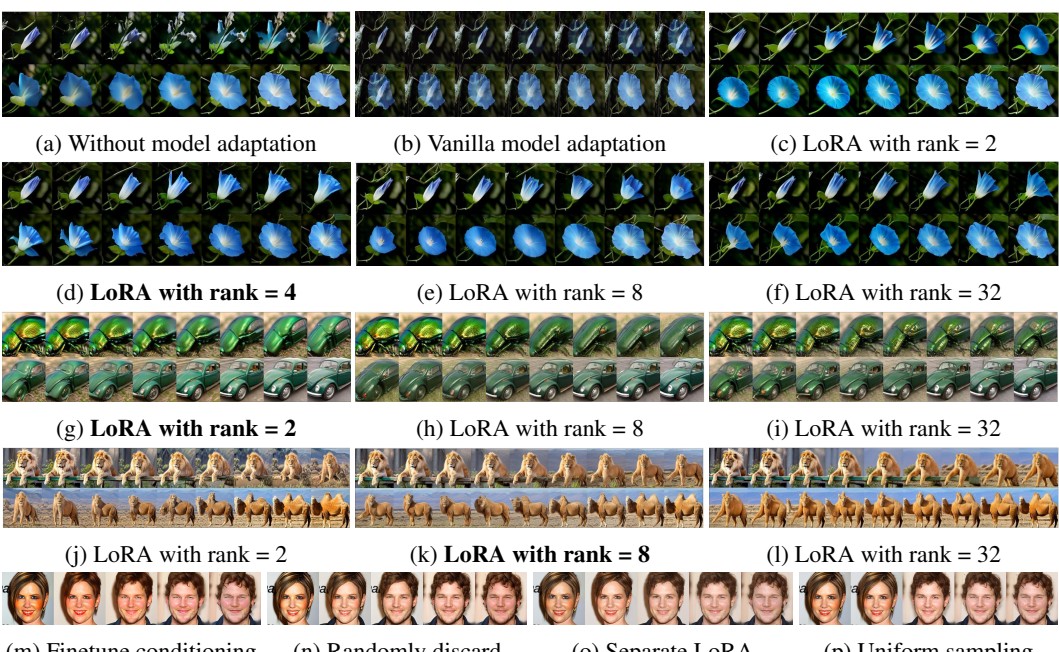

(a) Without model adaptation    (b) Vanilla model adaptation    (c) LoRA with rank = 2

(d) **LoRA with rank = 4**    (e) LoRA with rank = 8    (f) LoRA with rank = 32

(g) **LoRA with rank = 2**    (h) LoRA with rank = 8    (i) LoRA with rank = 32

(j) LoRA with rank = 2    (k) **LoRA with rank = 8**    (l) LoRA with rank = 32

(m) Finetune conditioning    (n) Randomly discard    (o) Separate LoRA    (p) Uniform sampling

Figure 4: Ablation studies under LoRA rank, unconditional noise estimates, and sampling methods. **LoRA rank:** (a)-(f) intra-class on flowers with $\gamma = 0.44$, $d_{\text{CLIP}} = 0.15$, (g)-(i) inter-class on beetle and car with $\gamma = 0.39$, $d_{\text{CLIP}} = 0.37$ , (j)-(l) inter-class on lion and camel with $\gamma = 0.49$, $d_{\text{CLIP}} = 0.29$. **Unconditional noise estimates:** (m) only finetune on conditioning, (n) randomly discard conditioning during finetune, (o) separate LoRA for unconditional estimates. **Sampling methods:** (o)-(p) perceptually uniform sampling versus uniform space sampling.

**Ablation Study on Model Adaptation.** We further ablate model adaption and show results on CelebA-HQ, AFHQ, and LSUN in Table 3 with heuristic LoRA rank for the bottleneck constraint and textual inversion. Since AFHQ and Celeba-HQ have different settings (female↔male, dog↔dog, etc.), we report the mean and standard deviation of all settings, see Appendix F for details. Overall, model adaptation leads to more direct morphing results (smaller LPIPS$_T$ and LPIPS$_M$) and realistic images (lower FID) and lower reconstruction error at endpoints. However, the higher PPL values reflect that uniform sampling may not work well with model adaptation.

We show performance w.r.t. LoRA ranks qualitatively in Figure 4, where the selected ranks by our heuristic bottleneck constraint are in bold. From the results, the ideal rank differs for different image pairs, and our proposed rPPD can select the rank with a qualified trade-off between image quality and directness (we also show $d_{\text{CLIP}} : 1 - \text{cosine\_similarity}$ as cosine distance in CLIP image embedding space for comparison). Quantitative analysis of LoRA rank on subsets of the benchmark datasets (female↔male, cat↔dog, church↔church) is reported in Table 2. Three settings of LoRA rank are considered which are: 1) heuristic rank with bottleneck constraint, 2) rank 4, and 3) rank 32. We observe that higher LoRA ranks have better quantitative results on the benchmark dataset which is different from the conclusion we have from the internet-collected images (see Figure 4(a)-(l)). We conjecture that the scope and diversity of benchmark datasets are more limited than the internet images; thus, phenomenon such as overly blurred interpolated images occurs less likely, resulting

in better performance on high LoRA ranks. Another observation is that reconstruction error at the endpoints does not decrease as the LoRA rank increases. We conjecture it is caused by the numerical error during the ODE inverse. We leave the study of these phenomena as our future work.

**Ablation Studies on Unconditional Noise Estimates and Perceptually Uniform Sampling.** We demonstrate the effects of different unconditional score estimate settings and sampling methods in Figure 4(m)-(p). If model adaptation only applies to conditional estimates, the inferior update of the unconditional estimates (due to sharing of parameters) results in a color bias, see Figure 4(m). Hence, we consider both randomly discard conditioning (Ho & Salimans, 2022) and apply a separate LoRA for unconditional score estimates.

In our experiments, applying a separate LoRA for unconditional score estimates leads to slightly higher image quality and is more robust to various guidance scales than randomly discard conditioning, see Figure 4(n)-(o) and Appendix E.6 for more details. We show an example of interpolation result from uniform sampling in Figure 4(p). Compared with Figure 4(o), transitions between images in Figure 4(p) are less seamless, valid our sampling technique.

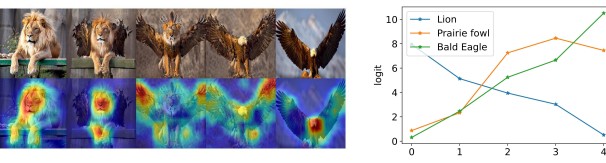

(a) GradCAM (morphing sequence)  (b) Logits (x: frame index)

Figure 5: Model explainability with IMPUS (ResNet50).

### 5.3 APPLICATION OF IMPUS TO OTHER TASKS

**Novel Sample Generation and Model Evaluation.** Given images from two different categories, our morphing method is able to generate a sequence of meaningful and novel interpolated images. Figure 5(a) shows a morphing sequence from a lion to an eagle. The interpolated images are creative and meaningful, which shows great potential in *data augmentation*. Our morphing technique can also be used for model evaluation and explainability. For example, the $4^{th}$ image of the morphing sequence looks like an eagle, but the highest logit (Figure 5(b)) is prairie fowl. With GradCAM (Selvaraju et al., 2017), the background seems distinguish the prairie fowl from the eagle.

**Video Interpolation.** Our morphing tools can also be directly applied for video interpolation. In Figure 6, given the start and end frames, we can generate realistic interpolated frames. We claim that more interpolated images can further boost our performance for video interpolation.

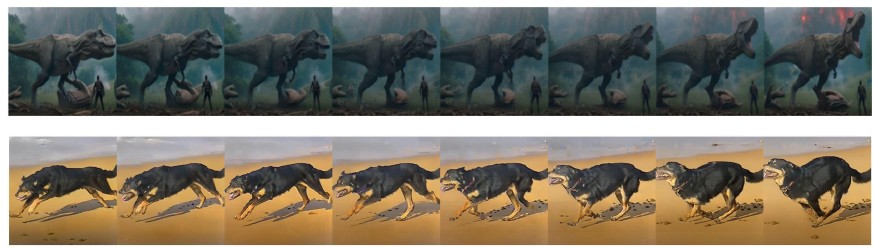

Figure 6: Video interpolation of Jurassic World (Bayona, 2018) and EpicSlowMotionDog (2013).

### 6 CONCLUSION

In this work, we introduce a diffusion-based image morphing approach with perceptually-uniform sampling, namely IMPUS, to generate smooth, realistic, and direct interpolated images given an image pair. We show that our model adaptation with a heuristic bottleneck constraint can suppress inferior transition variations and increase the image diversity with a specific directness. Our experiments validate that our approach can generate high-quality morphs and shows its potential for data augmentation, model explainability and video interpolation. Although IMPUS achieves high-quality morphing results, abrupt transitions occur when the given images have a significant semantic difference. In some contexts, our approach generates incorrect hands as well (see Appendix E.8). Hence, more investigations are desirable in our future work.

ACKNOWLEDGMENTS

This research is supported by the DARPA Geometries of Learning (GoL) program under the agreement No. HR00112290075 and the DARPA Environment-driven Conceptual Learning (ECOLE) program under agreement No. HR00112390061. The views, opinions, and/or findings expressed are those of the author(s) and should not be interpreted as representing the official views or policies of the Department of Defense or the U.S. government.

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

## APPENDIX

## A  PRELIMINARY ON DIFFUSION MODELS

**Denoising Diffusion Implicit Models (DDIM).**    In this paper, we use a DDIM (Song et al., 2021a).

For a sample $\mathbf{x}_t$, the reverse step is defined as follows with a variable $\sigma_t$

$$\mathbf{x}_{t-\Delta t} = \sqrt{\beta_{t-\Delta t}} \underbrace{\left( \frac{\mathbf{x}_t - \sqrt{1 - \beta_t} \epsilon_\theta^{(t)}(\mathbf{x}_t)}{\sqrt{\beta_t}} \right)}_{\text{``predicted } \mathbf{x}_0\text{''}} + \underbrace{\sqrt{1 - \beta_{t-\Delta t} - \sigma_t^2} \cdot \epsilon_\theta^{(t)}(\mathbf{x}_t)}_{\text{``direction pointing to } \mathbf{x}_t\text{''}} + \underbrace{\sigma_t \epsilon_t}_{\text{``random noise''}} .$$

(11)

By setting the free variable $\sigma_t$ in Eq. (11) as 0, we have a deterministic generative pass as follows

$$\frac{\mathbf{x}_{t-\Delta t}}{\sqrt{\beta_{t-\Delta t}}} = \frac{\mathbf{x}_t}{\sqrt{\beta_t}} + \left( \sqrt{\frac{1 - \beta_{t-\Delta t}}{\beta_{t-\Delta t}}} - \sqrt{\frac{1 - \beta_t}{\beta_t}} \right) \epsilon_\theta^{(t)}(\mathbf{x}_t) ,$$

(12)

which can be seen as the discretization of a continuous-time ODE (Song et al., 2021a)

$$\lim_{\Delta t \to 0} \frac{\mathbf{x}_t}{\sqrt{\beta_t}} - \frac{\mathbf{x}_{t-\Delta t}}{\sqrt{\beta_{t-\Delta t}}} = \lim_{\Delta t \to 0} \left( \sqrt{\frac{1 - \beta_t}{\beta_t}} - \sqrt{\frac{1 - \beta_{t-\Delta t}}{\beta_{t-\Delta t}}} \right) \epsilon_\theta^{(t)}(\mathbf{x}_t) ,$$

$$\mathrm{d}\overline{\mathbf{x}}(t) = \epsilon_\theta^{(t)} \left( \frac{\overline{\mathbf{x}}(t)}{\sqrt{\sigma^2(t) + 1}} \right) \mathrm{d}\sigma(t) ,$$

(13)

where $\overline{\mathbf{x}}(t) = \frac{\mathbf{x}_t}{\sqrt{\beta_t}}$ and $\sigma(t) = \sqrt{\frac{1 - \beta_t}{\beta_t}}$.

## B  FURTHER DISCUSSION ON COMPARISON WITH EXISTING METHODS

### B.1  COMPARISON WITH WANG & GOLLAND (2023)

We find the following major differences of Wang & Golland (2023) with our work: 1) task definition - Wang & Golland (2023) focus on *interpolation and novel image generation* but not directness, which is an important factor for image morphing. An indirect morph could generate a sequence of intermediate images irrelevant to the end point images (see Fig. 8), which also restricts their downstream applications. *Directness* of our method can lead to many downstream benefits, e.g., for video interpolation, it facilitates temporal coherency and consistency; for interpolation of an image and it's edited version, it can align the image transition with the target editing direction allowing intuitive user control over the magnitude of change; for data augmentation, it limits the image variations to be related to the endpoint images, making the interpolated images to be in-distribution; 2) we are

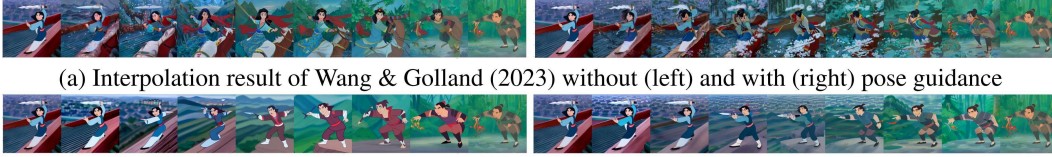

(a) Interpolation result of Wang & Golland (2023) without (left) and with (right) pose guidance

(b) Our image morphing results with generic text prompts (left) and model adaptation (right)

Figure 7: Compared with Wang & Golland (2023), our approach does not require extra guidance signals to achieve consistency in pose variation nor fine-grained text prompts. Together with the proposed model adaptation strategy, our approach demonstrates superior image morphing performance.

the first to propose that model adaptation is the key component for morphing. Despite its simplicity, this technique has been shown in our work to provide extraordinary improvements in both sample fidelity and directness. Meanwhile, unlike Wang & Golland (2023), the superior performance does not depend on careful selection for prompts (e.g., for boundary samples in Fig. 7, Wang & Golland (2023) uses two detailed prompts: *<HDR photo of a person practicing martial arts, crisp, smooth, photorealistic, high-quality wallpaper, ultra HD, detailed>* and *<HDR photo of a person leaning over, crisp, smooth, high-quality wallpaper>*, while we use a coarse prompt *<a photo of a cartoonish character>*), hyperparameters or random seeds. As shown in Fig. 7, it also enables our method to morph inputs with pose inconsistency, while Wang & Golland (2023) relies on an auxiliary pose estimation model to handle this problem; 3) The flexibility and robust performance shows the potential of our method to be adopted to a wide range of applications, e.g., in Sec. 5.3 of the main context we show that our work can be adopted for model evaluation and video interpolation. We provide further examples in Fig. 12 showing that our work can be seamlessly incorporated with an existing state-of-the-art image editing & translation pipeline (Brooks et al., 2023) that enables users to control the magnitude of change. Wang & Golland (2023) does not have such flexibility; 4) We propose perceptually uniform sampling while Wang & Golland (2023) uses standard uniform sampling. This has a significant impact on the quality of the results;

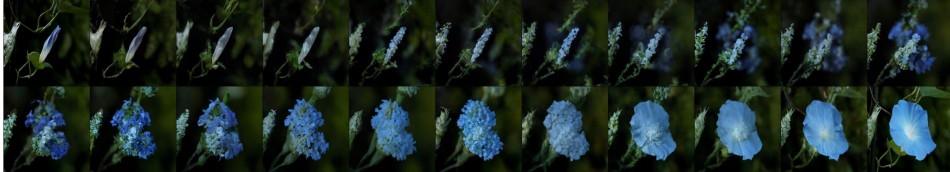

Figure 8: Indirect Interpolation from Wang & Golland (2023) generates irrelevant intermediate images which restrict application of downstream tasks.

We note that, in the original paper of Wang & Golland (2023), 26 pairs of images are claimed to be used, but there is access to only 25 pairs as provided in their public Repository `https://github.com/clintonjwang/ControlNet`. We present additional comparison with Wang & Golland (2023) in Fig. 9

## B.2 COMPARISON WITH OTHER MORPHING METHODS

We compare our morphing method with following open source automatic image morphing methods: DiffMorph (Nri, 2022), Automatic Image Morphing (Jankovics, 2020) (adapted from Python Image Morpher (Dowd, 2022)) as well as Neural Crossbreed (Park et al., 2020) and show the examples in Fig. 10. Automatic Image Morphing is non-learning based, DiffMorph is optimization-based while Neural Crossbreed is GAN-based. Since Neural Crossbreed requires image pairs relevant to the GAN training data, we only compare with them on the image pair provided in their paper. According to Fig. 10(a)-(c), morphing sequences generated by our method are smoother and more realistic compared with DiffMorph and Automatic Image Morphing. In Fig. 10(d), we show a failure case from Neural Crossbreed where the GAN model fails due to lack of training data. In summary, existing automatic morphing tools either create blurred and unrealistic intermediate images or cannot generalize to unseen data.

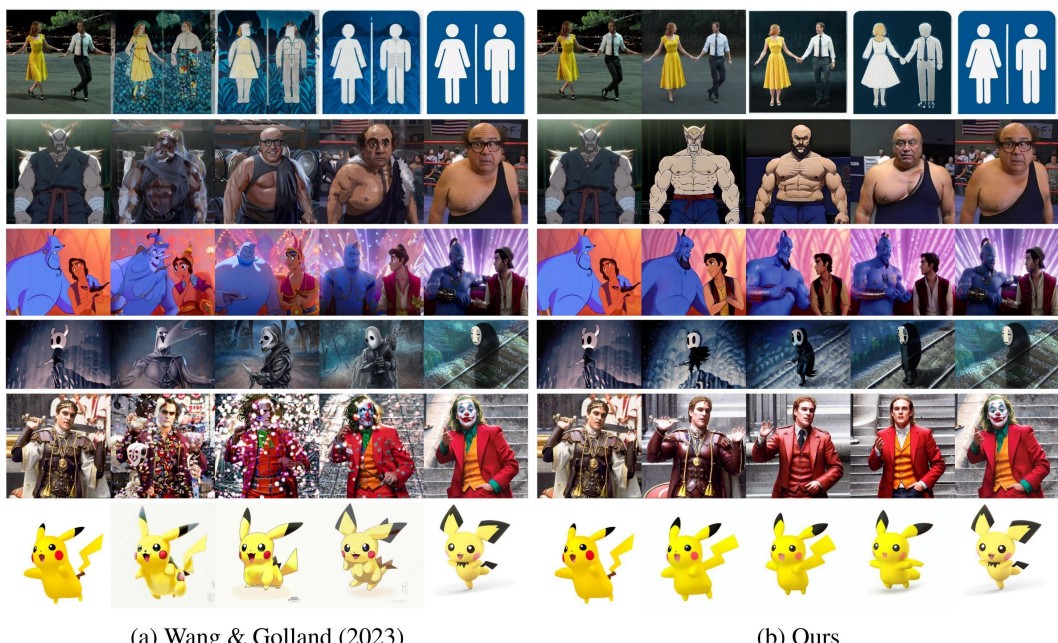

(a) Wang & Golland (2023)                    (b) Ours

Figure 9: Additional comparison results with Wang & Golland (2023). We rerun the code from Wang & Golland (2023) to generate 5-image sequences for comparison.

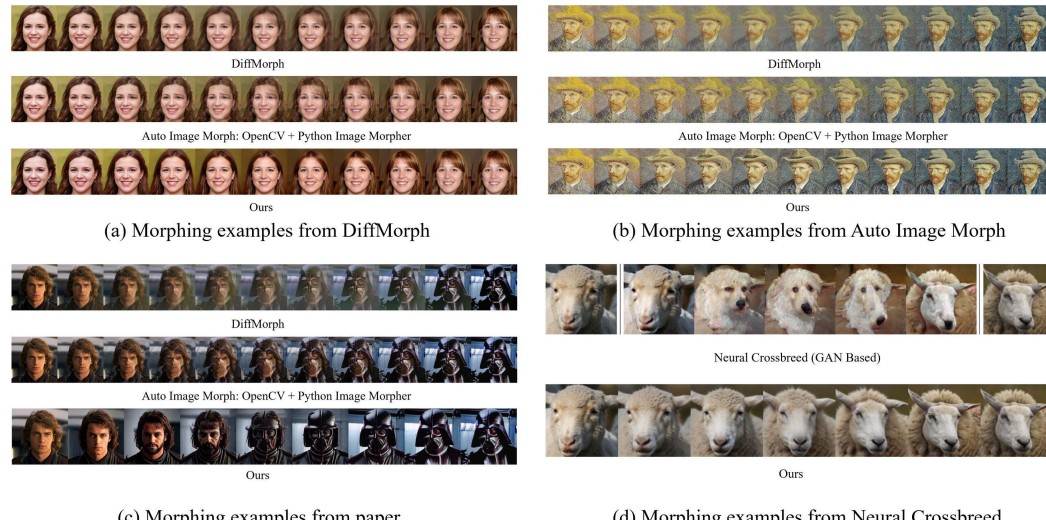

(a) Morphing examples from DiffMorph          (b) Morphing examples from Auto Image Morph

(c) Morphing examples from paper          (d) Morphing examples from Neural Crossbreed

Figure 10: Comparison between different open source implementations for automatic morphing. The image pair in (a) collected from DiffMorph. The image pair in (b) collected from Automatic Image Morphing. The image pair in (c) collected from Wang & Golland (2023). The image pair in (d) collected from Park et al. (2020). Overall, existing automatic morphing tools either create blurred and unrealistic intermediate images or cannot generalize to unseen data.

## C    EFFECT OF PROPOSED COMPONENTS

In Fig. 11, we provide the results of progressively adding proposed components in our work to the baseline of a pre-trained Stable Diffusion model, which uses user-provided prompts and discards the textual inversion and model adaptation: 1) Baseline, 2) Textual Inversion, 3) Textual Inversion + Vanilla Adaptation, 4) Textual Inversion + LoRA Adaptation (rank=4), 5) Textual Inversion + LoRA Adaptation (heuristic rank), 6) Textual Inversion + LoRA Adaptation (heuristic rank) + Uncondi-

tional Bias Correction. We can see that the baseline model fails to reconstruct one of the endpoint images and produces low fidelity samples. Textual inversion ensures a faithful reconstruction of the input endpoints. Model adaptation with LoRA can improve the perceptual directness of morphing path without suffering from the severe artifacts of vanilla model adaptation. An adaptively selected rank using the proposed rPPD metric shows improved directness than a commonly used rank, avoiding unnatural factor variations, e.g., using rank 4, the brightness of the left man's shirt first increases then decreases, while when heuristic rank is used, the brightness varies naturally. Results with the proposed unconditioned bias correction strategy shows finer details (such as hands and clothes).

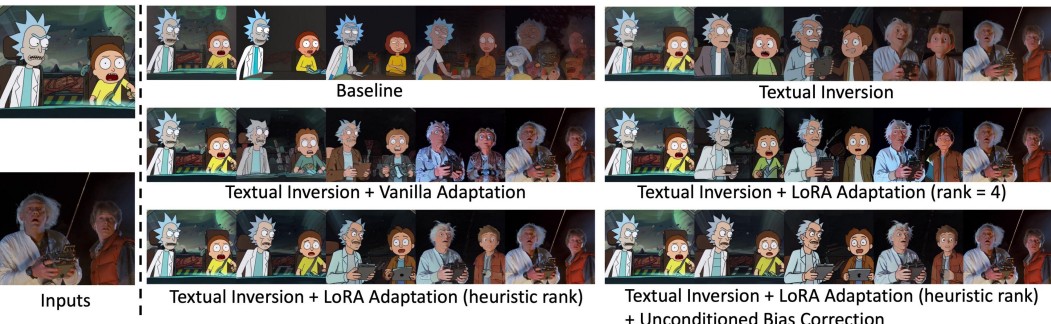

Figure 11: Results of progressively adding proposed components. The baseline is a pre-trained Stable-Diffusion which uses user-provided prompts and discards the textual inversion and modal adaptation process.

## D FURTHER IMPLEMENTATION DETAILS

For all the experiments, we use use a latent space diffusion model (Rombach et al., 2022), with pre-trained weights from Stable-Diffusion-v-1-4 [1]. Textual inversion is trained with AdamW optimizer (Loshchilov & Hutter, 2019), and the learning rate is set as 0.002 for 2500 steps. For the benchmark dataset, we perform text inversion for 1000 steps. LoRA is trained with Adam optimizer (Kingma & Ba, 2015), and the learning rate is set as 0.001. The hyperparameters of LPIPS is set to default settings.

### D.1 MODEL ADAPTATION AND PERCEPTUALLY UNIFORM SAMPLING

The overall process of our proposed framework is presented in Algorithm 1 and Algorithm 2 respectively.

## E JUSTIFICATION OF DESIGNS FOR EACH COMPONENT

In this section, we provide a detailed justifications for designating the proposed components and strategies of our image morphing method, including prompting & interpolation strategies, endpoint image selection details as well as model sampling & estimation strategies.

### E.1 COMMON PROMPT INITIALIZATION

Our text embedding interpolation strategy is based upon the local linearity of CLIP embedding space (Liang et al., 2022). To address the potential nonlinearity between the optimized embeddings of semantically distinct image pairs, we set a common prompt <*An image of* [*token*]> no matter the semantic gap between the input endpoint images, where for semantically distinct pairs, [*token*] is either a common root class (e.g., "animal") or a concatenation of the pair (e.g., "poker man"). Since two prompt embeddings are initialized with the same common prompt, they stay closed after optimization. We find that this simple method can avoid the non-linear and discontinuous issue in

---

[1]Source from https://huggingface.co/CompVis/stable-diffusion-v-1-4-original.

---

**Algorithm 1** Finetuning & inference process of IMPUS

---

**Require:** The pre-trained diffusion model $\epsilon_\theta$ with a text encoder $f_\phi$. Learning rates $\eta_1$, $\eta_2$. Input images $\mathbf{x}_0^{(0)}$ and $\mathbf{x}_0^{(1)}$. A common prompt $y$ for the endpoint images. Starting & ending interpolation scales $\alpha_{start}$, $\alpha_{end}$. Perceptual difference between consecutive sampling images $\delta_{lpips}$. Error tolerance for binary search $\epsilon_{tol}$. Number of steps for text inversion $T_{inv}$. Number of step for model adaptation $T_{adpt}$. Number of step for DDIM $T$.

1: // Phase 1: Textual Inversion (Sec.4.1)
2: **Initialize:** $e^{(0)} \leftarrow f_\phi(y)$, $e^{(1)} \leftarrow f_\phi(y)$
3: **for** $i \in \{1, ..., T_{inv}\}$ **do**
4:      Randomly sample time step $t$ and random noise $\epsilon_t \sim \mathcal{N}(0, \mathbf{I})$.
5:      Adding noise to data $\mathbf{x}_t^{(0)} \leftarrow \sqrt{\beta_t}\mathbf{x}_0^{(0)} + \sqrt{(1-\beta_t)}\epsilon_t$, $\mathbf{x}_t^{(1)} \leftarrow \sqrt{\beta_t}\mathbf{x}_0^{(1)} + \sqrt{(1-\beta_t)}\epsilon_t$
6:      $\mathbf{e}^{(0)} \leftarrow \mathbf{e}^{(0)} - \eta_1 \nabla_{\boldsymbol{e}^{(0)}} \mathcal{L}_{\mathrm{DPM}}(\mathbf{x}_0^{(0)}, \boldsymbol{e}^{(0)}; \theta)$
7:      $\mathbf{e}^{(1)} \leftarrow \mathbf{e}^{(1)} - \eta_1 \nabla_{\boldsymbol{e}^{(1)}} \mathcal{L}_{\mathrm{DPM}}(\mathbf{x}_0^{(1)}, \boldsymbol{e}^{(1)}; \theta)$
8: // Phase 2: Model Adaptation (Sec.4.2)
9: Calculating $\gamma$ according to Eq. 8.                  $\triangleright$ Heuristic rank selection
10: Setting LoRA rank $r_e \leftarrow 2^{\max(0, \lfloor 18\gamma - 6 \rfloor)}$
11: **for** $i \in \{1, ..., T_{adpt}\}$ **do**
12:      Low-rank model adaptation according to Eq. 7 and Eq. 9 with learning rate $\eta_2$.
13: // Phase 3: perceptually-uniform sampling
14: Obtaining initial latent states $\mathbf{x}_T^{(0)}$ and $\mathbf{x}_T^{(1)}$ via Eq. 2
15: $\boldsymbol{\alpha}_{list} \leftarrow \mathrm{Binarysearch}(\alpha_{start}, \alpha_{end}, \delta_{lpips}, \epsilon_{tol})$         $\triangleright$ Algorithm.2
16: **for** $\alpha$ in $\boldsymbol{\alpha}_{list}$ **do**                    $\triangleright$ Sampling process
17:      $\mathbf{e}^{(\alpha)} \leftarrow (1-\alpha)\mathbf{e}^{(0)} + \alpha\mathbf{e}^{(1)}$, $\mathbf{x}_T^{(\alpha)} \leftarrow \frac{\sin(1-\alpha)\omega}{\sin\omega}\mathbf{x}_T^{(0)} + \frac{\sin\alpha\omega}{\sin\omega}\mathbf{x}_T^{(1)}$    $\triangleright$ Interpolation
18:      **for** $t$ from $T$ to $1$ **do**
19:          $\mathbf{x}_{t-1}^{(\alpha)} \leftarrow \sqrt{\beta_{t-1}}\left( \frac{\mathbf{x}_t - \sqrt{1-\beta_t}\hat{\epsilon}_\theta^{(t)}(\mathbf{x}_t, \mathbf{e}^{(\alpha)})}{\sqrt{\beta_t}} \right) + \sqrt{1-\beta_{t-1}}\hat{\epsilon}_\theta^{(t)}(\mathbf{x}_t, \mathbf{e}^{(\alpha)})$
     **return** $\{\mathbf{x}_0^{(\alpha)}\}_{\alpha \in \boldsymbol{\alpha}_{list}}$

---

**Algorithm 2** Perceptually-Uniform Sampling with constant LPIPS

---

**Input:** $\alpha_{start}$: starting alpha value; $\alpha_{end}$: ending alpha value; $\delta_{lpips}$: perceptual difference between consecutive sampling images, $\epsilon_{tol}$: error tolerance for binary search
**Output:** $\boldsymbol{\alpha}_{list}$: list of alpha.

1: **procedure** BINARYSEARCH$(\alpha_{start}, \alpha_{end}, \delta_{lpips}, \epsilon_{tol})$
2:      $\boldsymbol{\alpha}_{list} \leftarrow [\alpha_{start}]$, $\alpha_{cur} \leftarrow \alpha_{start}$                    $\triangleright$ Initialization
3:      **while** $D(\alpha_{cur}, \alpha_{end}) > \delta$ **do**          $\triangleright$ Binary search of constant LPIPS
4:          $\alpha_{t1} \leftarrow \alpha_{cur}$, $\alpha_{t2} \leftarrow \alpha_{end}$, $\alpha_{mid} \leftarrow \frac{\alpha_{t1} + \alpha_{t2}}{2}$
5:          **while** $\left| \mathrm{LPIPS}(\mathbf{x}^{(\alpha_{cur})}, \mathbf{x}^{(\alpha_{mid})}) - \delta \right| > \epsilon_{tol}$ **do**
6:              **if** $\mathrm{LPIPS}(\mathbf{x}^{(\alpha_{cur})}, \mathbf{x}^{(\alpha_{mid})}) > \delta$ **then**
7:                  $\alpha_{t2} \leftarrow \frac{\alpha_{t1} + \alpha_{t2}}{2}$
8:              **else**
9:                  $\alpha_{t1} \leftarrow \frac{\alpha_{t1} + \alpha_{t2}}{2}$
10:              $\alpha_{mid} \leftarrow \frac{\alpha_{t1} + \alpha_{t2}}{2}$
11:          $\alpha_{cur} \leftarrow \alpha_{mid}$, $\boldsymbol{\alpha}_{list} \leftarrow \boldsymbol{\alpha}_{list} \bigcup \alpha_{cur}$        $\triangleright$ Append alpha to the list
12:      $\boldsymbol{\alpha}_{list} \leftarrow \boldsymbol{\alpha}_{list} \bigcup \alpha_{end}$                  $\triangleright$ Append the ending alpha
13:      **return** $\boldsymbol{\alpha}_{list}$

---

Editing with Instruct-pix2pix: "make the kid happy"

Magnitude of attribute editing controlled by IMPUS

Editing with Instruct-pix2pix: "put her in a bar"

Magnitude of environment transfer controlled by IMPUS

Editing with Instruct-pix2pix: "make it snowing"

Magnitude of style transfer controlled by IMPUS

Figure 12: Given a prompt-based image editing pipeline (Brooks et al., 2023), IMPUS can generate a gradual transition between source and edited images. The transitions generated by IMPUS is well-aligned with the target editing direction of the prompt, which can achieve intuitive user control over magnitude of change.

CLIP embedding space to some extent, where the usage of a common prompt is a crucial step which is illustrated in Fig. 13.

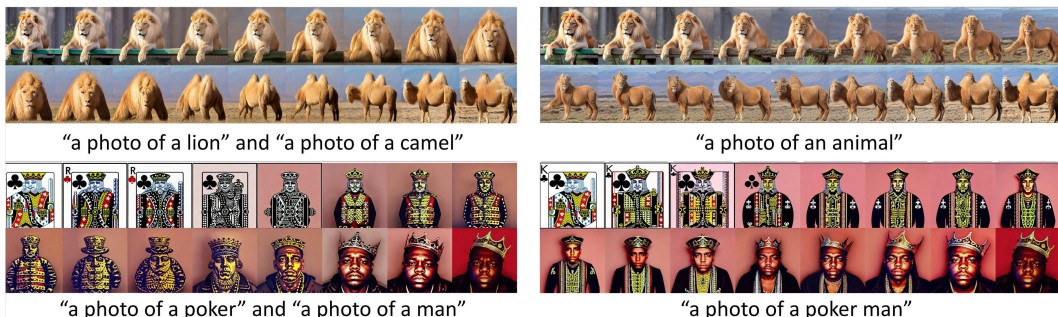

"a photo of a lion" and "a photo of a camel"          "a photo of an animal"

"a photo of a poker" and "a photo of a man"          "a photo of a poker man"

Figure 13: Given a pair of inter-class images, setting separate prompts lead to severe artifacts (top left) or drastic transitions (bottom left), suggesting a non-Euclidean property in CLIP embedding space. While our common prompt strategy (right) leads to natural transition between endpoints.

## E.2 INTERPOLATION IN LATENT STATE

The finding in Khrulkov et al. (2023) shows that $x_0$ and $x_T$ forms an Monge optimal transport in the L2 sense, this suggests that the similarity in latent state $x_T$ will also cause certain pixel-wise similarities between the generated image $x_0$. As shown in Figure 4 of Khrulkov et al. (2023), such similarity in latent state indeed reflect the structural information in generated images. We provide similar results in Fig. 14, where we fix the latent state $x_T$ for generations using a Stable Diffusion model with different text prompts, from where we can see significant alignment between the structure of generated images, which coincide with the finding of Khrulkov et al. (2023). This finding indicates that latent state $\mathbf{x}_T$ can reflect rich structural information about $x_0$, which makes

it essential to correctly interpolate $x_T$ in order to smoothly morph two images. We use spherical linear interpolation (slerp) instead of linear interpolation (lerp) to ensures that the interpolation can preserve the Euclidean norm of the interpolated latent state $\mathbf{x}_T$ (Huszár, 2017). Empirically, we find slerp works well while lerp often produces unnatural intermediate samples since the Euclidean norm of interpolated latent states are not preserved in lerp. See Fig. 15 for examples.

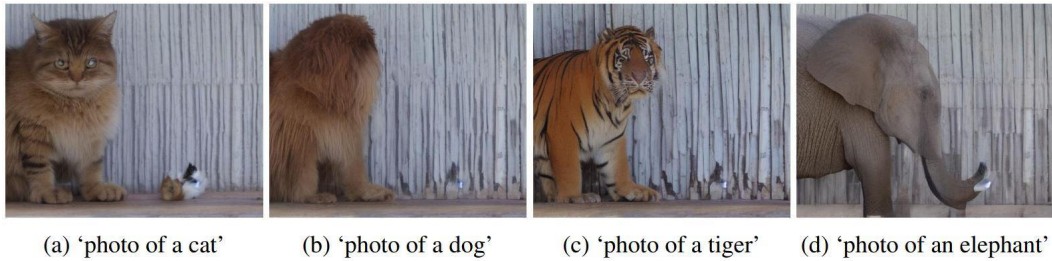

| (a) 'photo of a cat' | (b) 'photo of a dog' | (c) 'photo of a tiger' | (d) 'photo of an elephant' |

Figure 14: Using the same latent state $\mathbf{x}_T$, the generated images of Stable Diffusion using different prompts shows strong alignment in structural information.

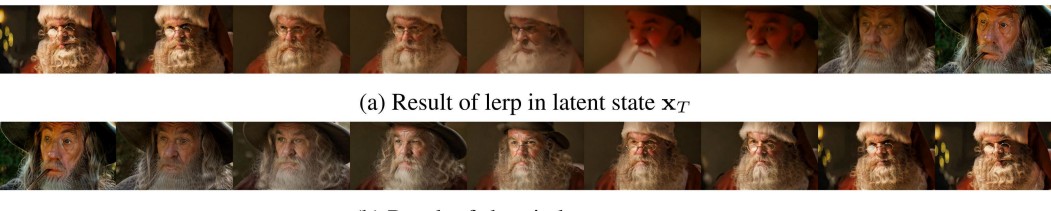

(a) Result of lerp in latent state $\mathbf{x}_T$

(b) Result of slerp in latent state $\mathbf{x}_T$

Figure 15: Linear interpolation (lerp) in latent state $\mathbf{x}_T$ produces unnatural intermediate samples. Samples with spherical linear interpolation (slerp) are natural-looking and consistent.

### E.3 SELECTION STRATEGY FOR ENDPOINT IMAGES

Image pairs selected in our works are collected from three sources: (1) Example image pairs provided in Wang & Golland (2023); (2) Additional images we collected from Internet; (3) Benchmark data for computer vision research. In terms of benchmark evaluation, we select the image pairs based on a simple protocol, i.e., for a given benchmark dataset, we randomly select image pairs whose similarity is under a certain LPIPS threshold (0.7 for cat and dog; 0.7 for cat↔dog; 0.7 for wild; 0.6 for female and male; 0.6 for female↔male; 0.55 for church). The intuition behind this process is that, the endpoint images should have certain perceptual connection for the interpolated images between them to be meaningful. In Fig. 16, we provide a counter example, showing that morphs between a pair of perceptually unrelated images can be meaningless.

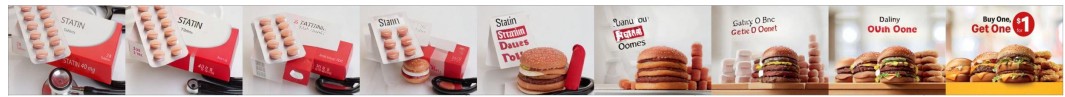

Figure 16: Morphing result between samples without perceptual connection may be meaningless.

### E.4 QUALITY BOOSTING

**Convex CFG Scheduling.** Although the interpolation between initial states and inverted textual embeddings can facilitate a smooth connection between inputs, however, we observe that the intermediate samples in the morphing path often show lower saturation than the ones near the boundaries. Intuitively, after model adaptation, the overall probability densities $p_\theta(\mathbf{x}_t, t)$ would concentrate more on the boundary samples, leading to lower density for regions in between. In order to provide extra guidance cues to guide the generation of intermediate samples via enforcing

a larger density ratio $p_\theta(\mathbf{x}_t, t|e)/p_\theta(\mathbf{x}_t, t)$, we propose a convex CFG scheduling in the form of $w_\alpha = w_{max} - 2(w_{max} - w_{min})|\alpha - 0.5|$, which can boost the generation quality for intermediate samples as shown in Fig. 17.

**Stochasticity Boosting.** For semantically distinct input pairs, artifacts may appear in details of the intermediate morphing results, where one possible cause is ODE sampler with limited time steps. As shown in prior works (Karras et al., 2022; Kwon et al., 2023), stochastic diffusion samplers often show improved quality than deterministic ones. Taking the reverse DDIM process in Eq. (11) as an example, a stochastic sampler sets a positive $\sigma_t$, which requires extra "denoising" by the diffusion model for newly introduced Gaussian noises on top of "denoising" for initial noise, which may also correct the prediction errors from earlier denoising steps. However, setting a positive $\sigma_t$ throughout the denoising process may fail to provide a faithful preservation of input content. To find a balance, using a total of 16-time steps, we find that setting $\sigma_t$ as 0 for the initial 6 steps and setting $\sigma_t$ as 1 for the rest 10 steps can faithfully retain the content from inputs while correcting some artifacts as shown in Fig. 17.

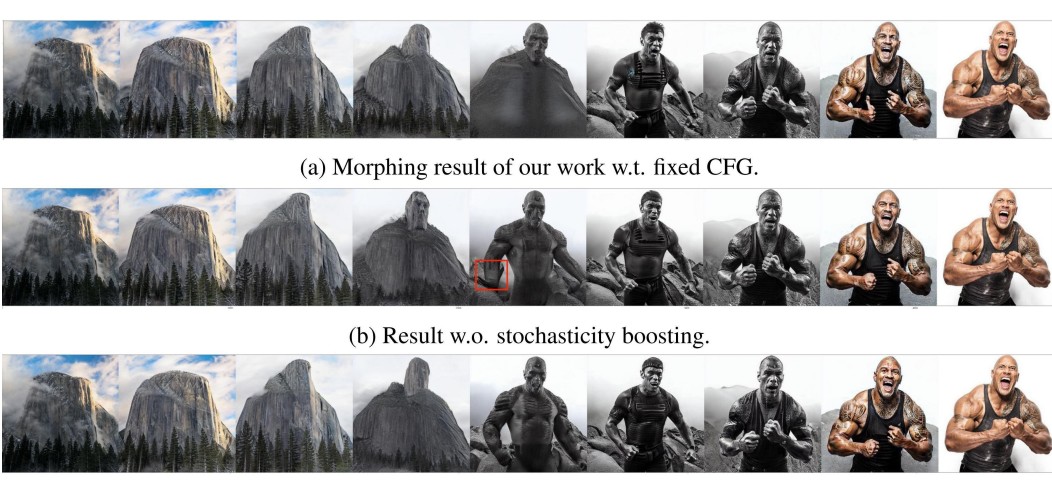

(a) Morphing result of our work w.t. fixed CFG.

(b) Result w.o. stochasticity boosting.

(c) Results of our w.t. quality boosting.

Figure 17: Illustration on the effect of quality boosting.

**Ablation Study on CFG.** We also study the effects of different hyperparameters, such as the scale of classifier-free guidance. We observe that inversion-only morphing requires a stronger CFG scale, while inversion with adaptation requires less guidance. In most situations, a large CFG scale can degrade the morphing quality and lead to a longer morphing sequence. We provide ablation studies of CFG on total LPIPS (Table. 4), Max LPIPS (Table. 5), FID (Table. 6) and PPL (Table. 7).

### E.5 FAILURE CASE WITHOUT TEXTUAL INVERSION

In our work, textual inversion is performed on the embedding of a coarsely initialized prompt. Alternatively, the optimized embedding could be replaced by a fine-grained human-designed text prompt. We find that using an appropriate prompt shared by two images can improve the smoothness of the morphing path without loss of image quality (see the first example in Fig. 18). However, for an arbitrary pair of images, it is usually difficult to choose such an ideal prompt (see the second example in Fig. 18 for failure cases where a human-labeled prompt is not informative enough to reconstruct given images, resulting in inferior performance). Instead, our textual inversion with a coarse initial prompt is more flexible and robust.

### E.6 FURTHER DETAILS OF SEPARATE LORA VS. RANDOMLY DISCARD CONDITIONING

During experiment, we observe that using separate LoRA for unconditional score estimation performs more stable (compared with randomly discard conditioning during finetune) for image inversion. For example, on CelebA (female-male pairs), maximum end point reconstructed error in

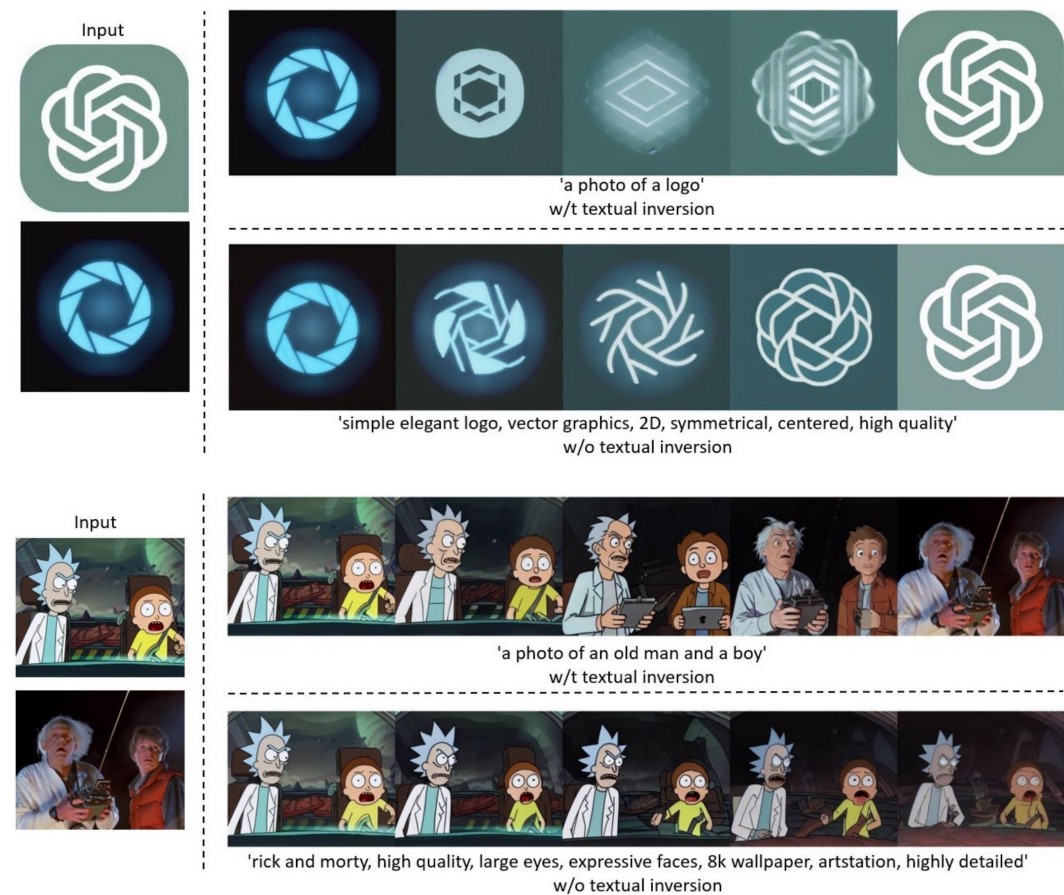

Figure 18: **Results w.o. textual inversion using detailed prompt**. In the first example where an appropriate description is selected to be shared by two images, the morphing path shows improved directness w.o. textual inversion. In the example below, an inappropriate prompt leads to an unfaithful reconstruction of inputs and downgraded quality.

terms of LPIPS is **0.137** for separate LoRA unconditional score estimates, the value for randomly discard conditioning during finetune is **0.211**. A poor reconstruction may lead to inferior quality; thus, we select separate LoRA for unconditional score estimation. We show two examples of reconstructed images in Fig. 19. To enhance the difference, we blend real as well as reconstructed images ($0.5 \times real + 0.5 \times recon.$) in Fig. 19(b). Other approaches could improve reconstruction quality as well, but applying separate LoRA for unconditional score estimates is simple yet effective enough for our task.

### E.7 INTUITION OF RELATIVE PERCEPTUAL PATH DIVERSITY

Fidelity and directness are two important criteria for our morphing task, but there is a trade-off between them: a blurred image sequence (generated from large rank) is likely to be more direct than a image sequence with high fidelity. If a pair of images has a large perceptual difference, a large rank adaptation could lead to blurred intermediate samples due to overfitting. The design of rPPD score is based on the trade-off between fidelity (higher quality morphs) and directness (smaller total perceptually change). **Directness**: The **accumulated LPIPS**, which measures the directness of preliminary morph, is proportional to the LoRA rank for adaptation. **Fidelity and Diversity**: The **perceptual difference of the endpoint images**, related to quality (blurriness, diversity, fidelity) of generated intermediate images after large rank adaptation, is inversely proportional to the LoRA rank for adaptation. Thus, we use the ratio to control the trade-off between the two, and a larger rPPD score indicates a larger LoRA rank can be applied to make the morphs more direct.

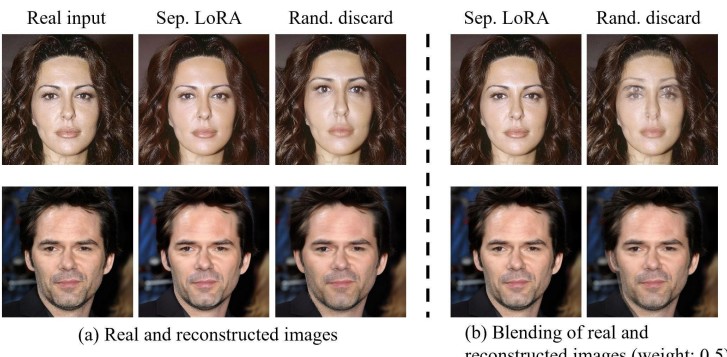

(a) Real and reconstructed images

(b) Blending of real and reconstructed images (weight: 0.5)

Figure 19: Comparison of reconstruction quality between (1) separate LoRA for unconditional score estimates and (2) randomly discard conditioning during finetune. The "ghost" effect from blending images indicated that randomly discard conditioning during finetune is less robust (compared with separate LoRA for unconditional score estimates) in terms of reconstruction.

## E.8 FAILURE CASES

Although IMPUS achieves high-quality morphing results, abrupt transitions occur when the given images have a significant semantic difference, see Figure 20 (a). In cases like Figure 20 (b), our approach generates incorrect hands, which is common for diffusion models.

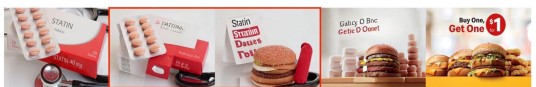 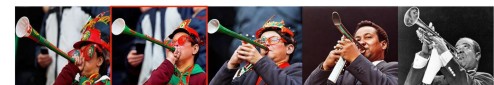

Figure 20: Failure cases of IMPUS

## F DETAILED QUANTITATIVE RESULTS

We provide detailed tables in this section. Different scales of guidance are considered: (1) CFG1: min:1.5 - max:2, (2) CFG2: min:2 - max:3, (3) CFG3: min:3 - max:4, and (4) CFG4: min:4 - max:5. Inv. represents text inversion and Adapt. represents model adaptation. The numbers reported in the main paper are the best numbers across all settings.

Table 4: Mean and standard deviation of total LPIPS.

| Dataset | Inv. Only CFG1 | Inv. Only CFG2 | Inv. Only CFG3 | Inv. Only CFG4 | Inv.+Adapt. CFG1 | Inv.+Adapt. CFG2 | Adapt. Only CFG1 | Adapt. Only CFG2 |
|---|---|---|---|---|---|---|---|---|
| male ↔ male | 1.38 ± 0.36 | 1.80 ± 0.66 | 1.51 ± 0.33 | 1.97 ± 0.40 | 1.02 ± 0.19 | 1.12 ± 0.20 | - | - |
| female ↔ female | 1.45 ± 0.53 | 1.63 ± 0.43 | 1.42 ± 0.28 | 1.84 ± 0.36 | 1.03 ± 0.16 | 1.16 ± 0.21 | - | - |
| female ↔ male | 1.50 ± 0.26 | 1.82 ± 0.38 | 1.60 ± 0.27 | 2.11 ± 0.30 | 1.16 ± 0.16 | 1.31 ± 0.18 | 1.13 ± 0.17 | 1.27 ± 0.18 |
| cat ↔ cat | 2.23 ± 0.62 | 2.70 ± 0.59 | 2.79 ± 0.81 | 3.39 ± 0.92 | 1.87 ± 0.44 | 2.18 ± 0.53 | - | - |
| dog ↔ dog | 2.59 ± 0.60 | 3.31 ± 0.77 | 3.06 ± 0.75 | 3.89 ± 1.03 | 2.20 ± 0.41 | 2.51 ± 0.49 | - | - |
| wild | 2.20 ± 0.40 | 2.70 ± 0.62 | 2.58 ± 0.60 | 3.23 ± 0.76 | 1.78 ± 0.31 | 2.06 ± 0.37 | - | - |
| cat ↔ dog | 2.04 ± 0.40 | 2.57 ± 0.42 | 2.62 ± 0.60 | 3.31 ± 0.86 | 1.75 ± 0.24 | 2.10 ± 0.40 | 1.61 ± 0.25 | 1.83 ± 0.26 |
| church ↔ church | 1.94 ± 0.44 | 2.58 ± 0.54 | 2.68 ± 0.56 | 3.76 ± 0.78 | 1.79 ± 0.34 | 2.26 ± 0.57 | - | - |

Table 5: Mean and standard deviation of max LPIPS.

| Dataset | Inv. Only CFG1 | Inv. Only CFG2 | Inv. Only CFG3 | Inv. Only CFG4 | Inv.+Adapt. CFG1 | Inv.+Adapt. CFG2 | Adapt. Only CFG1 | Adapt. Only CFG2 |
|---|---|---|---|---|---|---|---|---|
| male | 0.40 ± 0.06 | 0.43 ± 0.06 | 0.41 ± 0.05 | 0.43 ± 0.05 | 0.36 ± 0.05 | 0.38 ± 0.05 | - | - |
| female | 0.41 ± 0.06 | 0.43 ± 0.04 | 0.40 ± 0.05 | 0.43 ± 0.05 | 0.37 ± 0.04 | 0.38 ± 0.04 | - | - |
| female male | 0.42 ± 0.04 | 0.45 ± 0.03 | 0.42 ± 0.04 | 0.45 ± 0.04 | 0.39 ± 0.04 | 0.40 ± 0.03 | 0.37 ± 0.04 | 0.39 ± 0.04 |
| cat | 0.57 ± 0.06 | 0.60 ± 0.06 | 0.59 ± 0.07 | 0.61 ± 0.07 | 0.55 ± 0.07 | 0.56 ± 0.07 | - | - |
| dog | 0.59 ± 0.05 | 0.62 ± 0.04 | 0.61 ± 0.05 | 0.63 ± 0.04 | 0.59 ± 0.05 | 0.59 ± 0.05 | - | - |
| wild | 0.54 ± 0.04 | 0.56 ± 0.04 | 0.55 ± 0.04 | 0.58 ± 0.04 | 0.51 ± 0.04 | 0.51 ± 0.04 | - | - |
| cat dog | 0.53 ± 0.05 | 0.57 ± 0.04 | 0.56 ± 0.04 | 0.59 ± 0.04 | 0.52 ± 0.04 | 0.54 ± 0.03 | 0.50 ± 0.04 | 0.50 ± 0.03 |
| church | 0.40 ± 0.03 | 0.43 ± 0.04 | 0.44 ± 0.04 | 0.46 ± 0.04 | 0.42 ± 0.04 | 0.43 ± 0.04 | - | - |

Table 6: Mean and standard deviation of FID.

| Dataset | Inv. Only CFG1 | Inv. Only CFG2 | Inv. Only CFG3 | Inv. Only CFG4 | Inv.+Adapt. CFG1 | Inv.+Adapt. CFG2 | Adapt. Only CFG1 | Adapt. Only CFG2 |
|---|---|---|---|---|---|---|---|---|
| male | $57.58 \pm 0.32$ | $70.06 \pm 0.39$ | $63.26 \pm 0.36$ | $72.49 \pm 0.42$ | $46.19 \pm 0.25$ | $49.53 \pm 0.27$ | - | - |
| female | $47.17 \pm 0.29$ | $47.59 \pm 0.30$ | $46.66 \pm 0.31$ | $53.99 \pm 0.36$ | $36.39 \pm 0.23$ | $37.88 \pm 0.23$ | - | - |
| female male | $53.55 \pm 0.29$ | $61.39 \pm 0.35$ | $59.41 \pm 0.33$ | $67.82 \pm 0.38$ | $48.20 \pm 0.25$ | $48.69 \pm 0.26$ | $48.89 \pm 0.25$ | $50.92 \pm 0.26$ |
| cat | $36.53 \pm 0.26$ | $39.49 \pm 0.29$ | $40.75 \pm 0.30$ | $45.87 \pm 0.34$ | $32.09 \pm 0.21$ | $35.00 \pm 0.22$ | - | - |
| dog | $61.45 \pm 0.32$ | $72.59 \pm 0.38$ | $55.17 \pm 0.34$ | $61.73 \pm 0.39$ | $49.62 \pm 0.28$ | $50.15 \pm 0.28$ | - | - |
| wild | $23.69 \pm 0.20$ | $30.33 \pm 0.24$ | $23.28 \pm 0.22$ | $27.31 \pm 0.25$ | $19.61 \pm 0.17$ | $20.11 \pm 0.18$ | - | - |
| cat dog | $56.13 \pm 0.30$ | $63.61 \pm 0.36$ | $51.50 \pm 0.33$ | $56.46 \pm 0.38$ | $45.82 \pm 0.26$ | $49.12 \pm 0.29$ | $46.28 \pm 0.26$ | $48.57 \pm 0.29$ |
| church | $43.64 \pm 0.42$ | $44.61 \pm 0.44$ | $42.48 \pm 0.44$ | $45.35 \pm 0.50$ | $36.52 \pm 0.36$ | $38.87 \pm 0.39$ | - | - |

Table 7: Mean and standard deviation of PPL.

| Dataset | Inv. Only CFG1 | Inv. Only CFG2 | Inv. Only CFG3 | Inv. Only CFG4 | Inv.+Adapt. CFG1 | Inv.+Adapt. CFG2 | Adapt. Only CFG1 | Adapt. Only CFG2 |
|---|---|---|---|---|---|---|---|---|
| male | $187.80 \pm 464.17$ | $293.92 \pm 712.75$ | $583.94 \pm 1816.54$ | $877.37 \pm 2765.22$ | $187.10 \pm 423.31$ | $323.73 \pm 731.63$ | - | - |
| female | $149.92 \pm 275.70$ | $258.72 \pm 628.28$ | $505.10 \pm 1381.98$ | $591.45 \pm 2851.81$ | $179.65 \pm 375.86$ | $305.35 \pm 694.54$ | - | - |
| female male | $251.19 \pm 480.31$ | $421.73 \pm 1228.64$ | $901.66 \pm 4177.53$ | $980.52 \pm 3007.18$ | $238.87 \pm 557.78$ | $394.47 \pm 916.87$ | $248.06 \pm 766.20$ | $381.27 \pm 1616.40$ |
| cat | $432.71 \pm 823.13$ | $876.83 \pm 2735.96$ | $1451.21 \pm 3078.00$ | $1977.90 \pm 5408.94$ | $710.25 \pm 1600.53$ | $1196.90 \pm 2326.12$ | - | - |
| dog | $475.55 \pm 817.85$ | $1053.61 \pm 2423.10$ | $2005.83 \pm 4091.21$ | $2534.77 \pm 6712.03$ | $918.50 \pm 2246.47$ | $1674.60 \pm 4529.26$ | - | - |
| wild | $412.65 \pm 713.96$ | $747.22 \pm 1327.47$ | $1383.24 \pm 3071.43$ | $1395.23 \pm 3781.30$ | $605.59 \pm 1459.92$ | $1058.10 \pm 2516.31$ | - | - |
| cat dog | $335.48 \pm 605.63$ | $649.59 \pm 1206.66$ | $1233.78 \pm 2332.01$ | $1233.80 \pm 2580.24$ | $485.48 \pm 893.48$ | $905.98 \pm 2081.32$ | $451.85 \pm 916.24$ | $864.58 \pm 2048.68$ |
| church | $663.33 \pm 1256.84$ | $1632.14 \pm 3353.08$ | $4631.25 \pm 10567.48$ | $10460.97 \pm 26939.89$ | $1691.48 \pm 3804.02$ | $3823.07 \pm 9484.52$ | - | - |

## F.1 RATIONALE OF TOTAL LPIPS AS INDICATOR OF SMOOTHNESS FOR PERCEPTUALLY UNIFORM SAMPLING

We provide some rationale of how total LPIPS related to the (1) rate of change between consecutive images and (2) perceptual path length (PPL) here. Given two images $x^{(0)}$ and $x^{(1)}$, a diffusion model $x^{(\alpha_i)} = \mathcal{F}(\alpha_i, x^{(0)}, x^{(1)})$ generates an interpolated sample (between $x^{(0)}$ and $x^{(1)}$) based on the interpolation parameter $\alpha_i$. For a specified small delta LPIPS value $\Delta_{LPIPS}$ (which is almost imperceptible by human), perceptually uniform sampling ensures that $LPIPS(x^{(\alpha_i)}, x^{(\alpha_{i+1})}) = \Delta_{LPIPS}$ and the morphing process requires $N$ interpolated samples to start from $x^{(0)}$ to $x^{(1)}$. We define this set of $N$ (non-uniform) interpolation scale as $A = \{\alpha_1, ..., \alpha_N\}$, and a set of $N$ uniform interpolation scale as $B = \{\frac{1}{N+1}, \frac{2}{N+1}, ..., \frac{N}{N+1}\}$, then perceptually uniform sampling can be considered as a function $\mathcal{G} : B \to A$ such that $\mathcal{G}(\frac{i}{N+1}) = \alpha_i$, and the generation of interpolated samples (given perceptually uniform sampling) can be written as $x^{(\alpha_i)} = \mathcal{F}(\mathcal{G}(\beta_i), x^{(0)}, x^{(1)})$ where $\beta_i \in B$. Thus, **rate of change** $\frac{\Delta_{LPIPS}}{\Delta_\beta}$ is always $(N+1) \cdot \Delta_{LPIPS}$ which is the **total LPIPS** along the morphing sequence. One noteworthy observation is that $N$ and $\Delta_{LPIPS}$ are not independent: as $N$ increase, $\Delta_{LPIPS}$ will decrease. The intuition behind is that for a fixed number of $N$ interpolated samples (from $x^{(0)}$ to $x^{(1)}$), models with larger total LPIPS will have a larger rate of change (in term of LPIPS) between two consecutive samples (less smooth given fixed number of interpolated points). Another metric for smoothness is **PPL**: $\text{PPL}_\epsilon = \mathbb{E}_{\alpha \sim U(0,1)}[\frac{1}{\epsilon^2} LPIPS(x^{(\alpha)}, x^{(\alpha+\varepsilon)})]$. Following notations from the previous paragraph, if we let $\varepsilon = \frac{1}{N+1}$, PPL for perceptually uniform sampling is $\text{PPL}_\epsilon = \mathbb{E}_{\alpha \sim U(0,1)}[(N+1)^2 \cdot \Delta_{LPIPS}] = (N+1)^2 \cdot \Delta_{LPIPS} = \frac{(\text{total LPIPS})^2}{\Delta_{LPIPS}}$. Standard PPL calculation requires $\varepsilon = \frac{1}{N+1} \ll 1$. In terms of perceptually-uniform sampling, $N$ should be defined in a manner such that $\Delta_{LPIPS}$ are almost imperceptible by human. Using total LPIPS to approximate PPL may not be precise due to numerical approximation, but the trend should be similar (larger total LPIPS indicates larger PPL).

## F.2 IMPACTS OF CLASSIFIER-FREE GUIDANCE (CFG) SCALES

We discover that, given fixed LPIPS difference for perceptually-uniform sampling, larger CFG scales are likely to generate longer sequences (less direct morphs). The reconstruction quality also decrease for large CFG scales, but overall the differences are not visually significant. We show an example of sequence generated from different CFG scale in Figure 21. Detailed tables in our appendix as well.

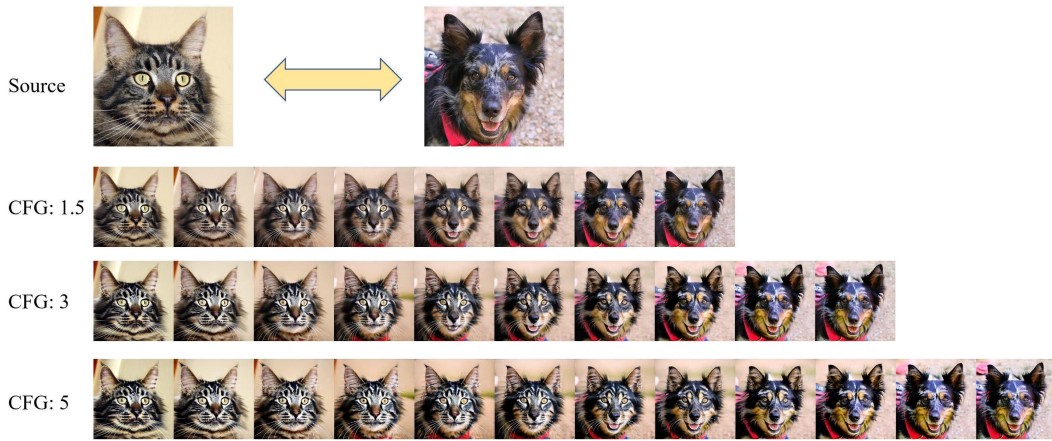

Figure 21: Perceptually-Uniform Sampling with different CFG scales. The difference of LPIPS between consecutive images are set to 0.2.

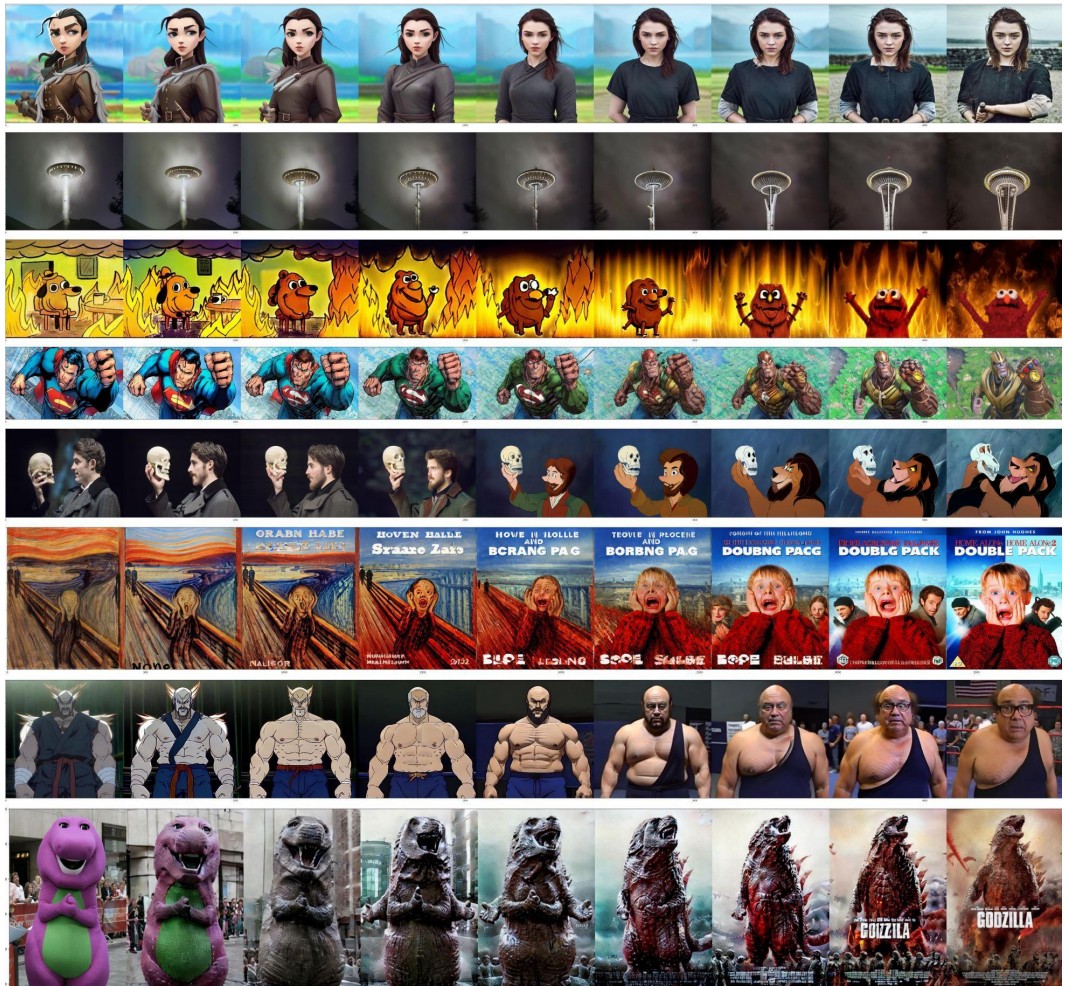

Figure 22: Additional morphing results with 9 frames.

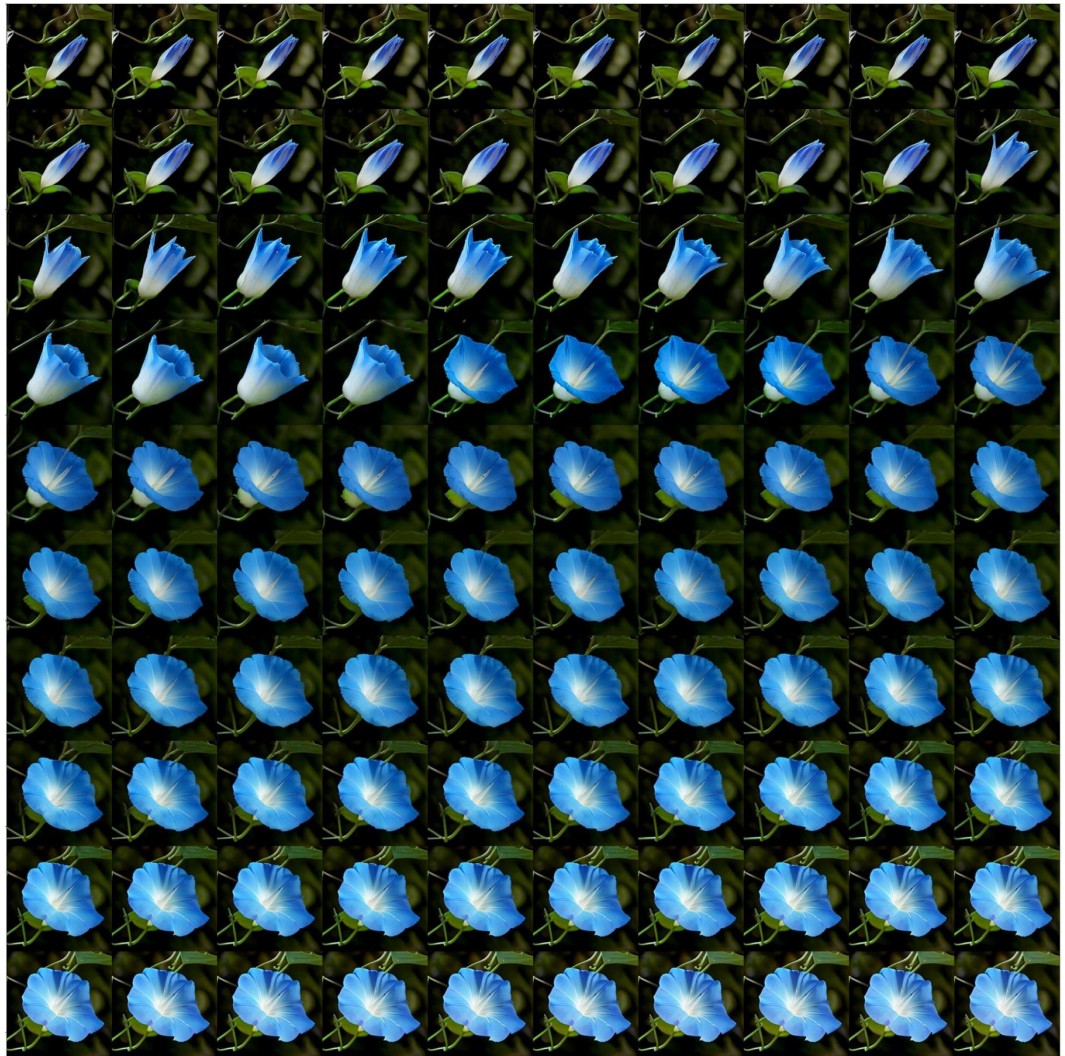

Figure 23: 100 interpolated images of "flower".

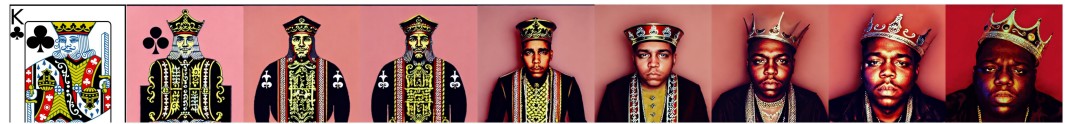

Figure 24: Poker card to a human.

## G  ADDITIONAL MORPHING RESULTS AND DOWNSTREAM APPLICATIONS

**Long Trajectory Morphing Results**   To better illustrate the smoothness of morphing with our proposed IMPUS, we illustrate examples of 9 interpolated images in Fig. 22. We also illustrate 100 interpolated images using our uniform sampling in Fig. 23

**More Examples of Downstream Applications**   We show some additional examples of downstream applications in Fig. 25-27

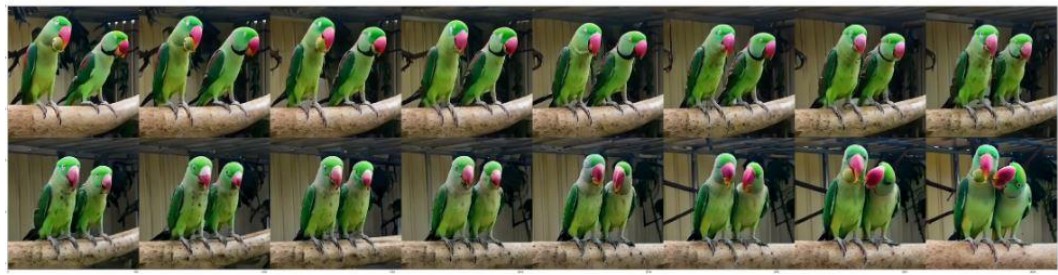

Figure 25: Interpolation of two video frames - teleporting parrot.

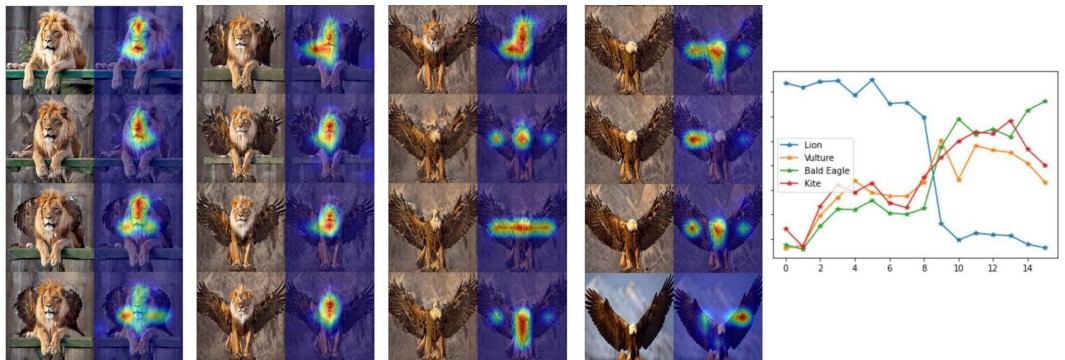

Figure 26: Model explainability with our morphing tool (SwinT).

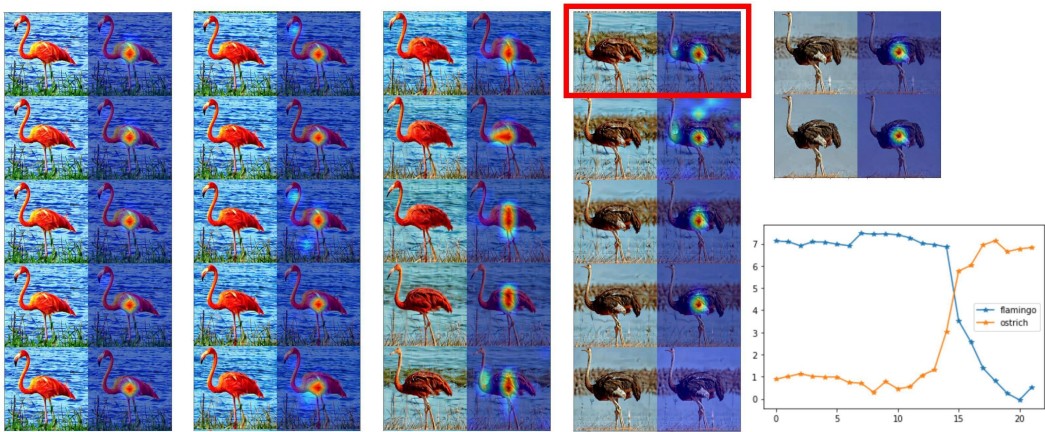

Figure 27: Model explainability with our morphing tool (SwinT).

## H  IMAGE ATTRIBUTION

- 25 pairs of images: `https://github.com/clintonjwang/ControlNet`

- Flower: `https://www.youtube.com/watch?v=-sOdrJ6pusA`
- Beetle:`https://lancasteronline.com/features/home_garden/`
  `antiques-market-vintage-beetle-is-your-chance-to-relive-\`
  `the-60s/article_308b7156-6713-11e7-95ed-732b1e8d88de.html`
- Car: `https://thebugagenda.com/green-iridescent-beetles/`
- Camel:`https://www.balisafarimarinepark.com/`
  `what-makes-camel-became-a-unique-animal/`
- Lion:`https://www.sfzoo.org/african-lion/`
- Eagle:`https://www.pinterest.com/pin/411516484700344157/`
- Ostrich:`https://www.newscientist.com/article/`
  `2313490-ostrich-necks-act-as-a-radiator-to-control-\`
  `their-head-temperature/`
- Flamingo:`https://www.puertoricodaytrips.com/`
  `pinky-the-flamingo/`
- Movie clip of Jurassic World: `https://www.imdb.com/title/tt4881806/`
- Running dog: `https://youtu.be/_caOk6lycb0?si=g5o_j27IWcQWSKQp`
- Parrot: `https://www.youtube.com/shorts/HJG1laf9Sq8?`
- Movie poster Barbie: `https://www.imdb.com/title/tt1517268/`
- Movie poster Oppenheimer:`https://www.imdb.com/title/tt15398776/`

