# OpenReview forum: "IMPUS: Image Morphing with Perceptually-Uniform Sampling Using Diffusion Models"
_ICLR.cc/2024/Conference — ICLR 2024 poster_

### Official Review · Reviewer_UYJ7 · 2023-10-28

**Soundness:** 2 fair
**Presentation:** 2 fair
**Contribution:** 2 fair
**Rating:** 6
**Confidence:** 3

**Summary:**

This paper presents a diffusion-based image morphing approach aiming to produce smooth, direct, and realistic interpolations given an image pair.  The authors interpolate in the locally linear and continuous text embedding space and Gaussian latent space. An "adaptive" bottleneck constraint based on a relative perceptual path diversity score that automatically controls the bottleneck size and balances the diversity along the path with its directness. Experiments verify the effectiveness to some extent. Though the method is clearly described, there exist some issues with both motivations and experiments.

**Strengths:**

* Image morphing in computer graphics is useful. Image morphing of nature images seems visually interesting to me.
* The method of image morphing with perceptually-uniform sampling using diffusion models is clearly described.
* The paper is OK to read.

**Weaknesses:**

* The motivation for image morphing of conceptually different nature images is not clear to me. Why should we study the image morphing of natural images without any human involvement? To my understanding, it may be reasonable to morph images for a certain purpose, e.g., continuously changing the poses or colors, via human involvement. However, how can we guarantee the generated images fit our purpose without any human involvement?
* The perceptually-uniform sampling is not reasonable to me. If I am right, morphing should be that given a uniform sequence in time-space, e.g., [0,1], and generating images "uniformly" changing between start and end images. In perceptually-uniform sampling, can we obtain the images corresponding to a given middle time?
* The authors argue "adaptive" bottleneck constraint. However, I can not see how the rank is adaptively learned.
* More discussion on the perceptual path diversity is encouraged. For example, why should we use this to measure the diversity? Why use the formulation in Eq. (8)? Does large $\gamma$ imply higher diversity?
* It would be better to provide an algorithm for both fine-tuning and inference.
* To show the superiority of the proposed method, more baseline approaches should be compared in experiments, such as the image morphing approaches in the related works.

**Questions:**

* In Eq. (5), $\nu$ is a distribution, why it should belong to the data manifold $\mathcal{M}$ which should be the set of samples?
* The embedding space of CLIP is locally linear. Since the two semantically distinct text embeddings are not in the global linear space, does it contradict the linear interpolation between the optimized embedding spaces?
* What is the logic for "the diffusion process between $\bf x_0$ and ${\bf x}_T$ can be seen as an optimal transport mapping, therefore, we apply spherical linear interpolation"?

---

> ### Author Response · Authors · 2023-11-17
> **Response to reviewer UYJ7 - Part 1/2**
>
> Thanks for your constructive feedbacks. Following are some responses to your questions.
>
> >**W1: The motivation for image morphing of conceptually different nature images, Why should we study the image morphing of natural images without any human involvement? It may be reasonable to morph images for a certain purpose, e.g., continuously changing the poses or colors, via human involvement. However, how can we guarantee the generated images fit our purpose without any human involvement?**
>
> Thank you for your thought-provoking question. There is actually one *crucial element of human involvement*, namely the **selection of the endpoint images**. Direct, smooth and realistic image morphing can be used for different downstream purposes depending on the choice of selected images.
>
> If the selected endpoints directly reflect a certain factor variation, e.g., an object attribute, pose or style, then our method will provide a sequence where this factor is varied continuously, since other variations would hinder the directness of the path. This can be highly beneficial for numerous applications. For example, in controllable generation, as shown in Fig.6 of the revised appendix, we integrate our method with an existing image editing pipeline [1], where our method generates a gradual transition between the original image and its edited version according to certain prompt. With our proposed model adaptation strategy, the transition can reveal a direct & high-fidelity variation that aligns with the editing direction. Thus we believe **our work can also contribute to controllable generation applications in terms of enabling users to control the magnitude of editing or translation**.
>
> Morphing two conceptually different images also has many important usages:
> * **Novel content creation:** Using image morphing techniques to create novel content has been studied for many years [2]. For large-scale generative models, one of the important purposes is to achieve creation of customized novel content that is difficult to see in real world or would require specialized and extensive human involvement [3]. There are two major solutions to achieve novel content generation: a) to design a compositional generative model [4], i.e., combining visual concepts or attributes that exist in a training dataset to generate novel examples, which has been studied by numerous existing works works (e.g., [4,5,6]); and b) to design an interpolative generative model, e.g., the example of interpolating a "lion" and an "eagle" in our work. The former can exponentially expand the control space as more concepts are learned by the generative model, while the latter can potentially provide infinite use of learned concepts. Thus we believe that morphing between different concepts is a highly useful task, and our work is the one of the first attempts.
> * **Data augmentation:** As our proposed method can generate **perceptually direct transitions** between endpoint images, for a given dataset, the interpolated samples with our method could be considered as in-distribution images, which can be used as a powerful tool to achieve **semantically meaningful and diverse data augmentation**.
> * **Model robustness analysis:** Our work could also be used to generate conceptually-ambiguous samples (e.g., the morph between a "cat" and a "dog") to study the robustness and uncertainty awareness of computer vision models.

---

> ### Author Response · Authors · 2023-11-17
> **Response to reviewer UYJ7 - Part 2/2**
>
> >**W2: The perceptually-uniform sampling is not reasonable to me. If I am right, morphing should be that given a uniform sequence in time-space, e.g., [0,1], and generating images "uniformly" changing between start and end images. In perceptually-uniform sampling, can we obtain the images corresponding to a given middle time?**
>
> Perceptually-uniform sampling just finds a continuous mapping from $[0,1] \to [0,1]$ such that the perceptual gap between images $t$ and $t+\delta$ is roughly the same as that of images $t+\delta$ and $t+2\delta$. A middle time (0.5) is still meaningfully defined, and an image can be generated at this time.
> In this context, time-uniformity would only be more meaningful than perceptual-uniformity if the underlying space was uniform in some well-defined way.
>
> We provide feedbacks based on our understanding of this question, and more discussion is encouraged.
>
> >**W3: The authors argue "adaptive" bottleneck constraint. However, I can not see how the rank is adaptively learned.**
>
> The rank is adaptively calculated based on the input image pair.
>
> >**W4: More discussion on the perceptual path diversity is encouraged.**
>
> Please refer to general comment 3.
>
> >**W5: Provide an algorithm for both fine-tuning and inference.**
>
> Thanks for the suggestion. Please refer to Algorithm 1 in the revised appendix.
>
> >**W6: More baselines.**
>
> Please refer to general comment 1 for additional comparison study.
>
> >**Q1: In Eq. (5), $v$ is a distribution, why it should belong to the data manifold $\mathcal{M}$ which should be the set of samples.**
>
> Thanks for pointing out the typo. We define $\nu$ as probability measure on manifold $\mathcal{M}$.
>
> >**Q2: Since the two semantically distinct text embeddings are not in the global linear space, does it contradict the linear interpolation between the optimized embedding spaces?**
>
> Thanks for your concern. We avoid this problem by our textual inversion method. Firstly, we set a common prompt **<An image of [token]>** for the inputs no matter their semantic gap, where for semantically distinct pairs, [token] is either a common root class (e.g., "animal") or a concatenation of the pair (e.g., "poker man"). Secondly, we optimize the initial prompt embeddings w.r.t. to the input images. Since two prompt embeddings are initialized with the same common prompt, they stay closed after optimization. We find that this simple method can avoid the nonlinear \& discontinuous issue in CLIP embedding space to some extent, where the usage of a common prompt is a crucial step which is illustrated in Fig.7 of the revised appendix.
>
> >**Q3: What is the logic for "the diffusion process between $x_0$ and $x_T$ can be seen as an optimal transport mapping, therefore, we apply spherical linear interpolation"?**
>
> Please refer to general comment 2.
>
> **References:**
>
> [1] Tim Brooks, Aleksander Holynski, and Alexei A Efros. Instructpix2pix: Learning to follow image editing instructions. In Proceedings of the IEEE/CVF Conference on Computer Vision and Pattern Recognition, pp. 18392–18402, 2023.
>
> [2] Wolberg, George. "Image morphing: a survey." The visual computer 14.8-9 (1998): 360-372.
>
> [3] Nichol, Alex, et al. "Glide: Towards photorealistic image generation and editing with text-guided diffusion models." arXiv preprint arXiv:2112.10741 (2021).
>
> [4] Liu, Nan, et al. "Compositional visual generation with composable diffusion models." European Conference on Computer Vision. Cham: Springer Nature Switzerland, 2022.
>
> [5] Kumari, Nupur, et al. "Multi-concept customization of text-to-image diffusion." Proceedings of the IEEE/CVF Conference on Computer Vision and Pattern Recognition. 2023.
>
> [6] Nie, Weili, Arash Vahdat, and Anima Anandkumar. "Controllable and compositional generation with latent-space energy-based models." Advances in Neural Information Processing Systems 34 (2021): 13497-13510.

---

> ### Comment · Reviewer_UYJ7 · 2023-11-22
>
> I thank the authors for the response. Some of my concerns are addressed but others are not.
>
> Regarding the usefulness of natural image morphing, I am still not convinced that natural image morphing is important. I do believe that image morphing is useful in some applications. For example, in computer graphics, morphing between two facial expressions of one person is useful, which requires preserving the identity of the person. However, the importance of morphing between two natural images given only the start and end images is not clear. [1] and [3-6] studies controllable image generation or image editing rather than image morphing. "Novel content creation", "data augmentation", and "model robustness analysis" mainly increase the diversity of datasets, for which image morphing may not be appropriate. Because there seems to be no need for directness. I suggest that the authors focus on the morphing of certain images instead of arbitrary natural images.
>
> Regarding the adaptiveness of the rank, the paper empirically sets it using an empirically designed function. I would not say that it is adaptive because why to choose this formulation and how to set the hyper-parameters in the function are not clear.
>
> Meanwhile, a probability measure $\nu$ on $\mathcal{M}$ is function of a subset of $\mathcal{M}$, i.e., $\nu:\sigma(\mathcal{M})\rightarrow R$, where $\sigma(\mathcal{M})$ a set of subsets of $\mathcal{M}$ (called $\sigma$-algebra).

---

> ### Author Response · Authors · 2023-11-22
> **Follow Up Response to Reviewer UYJ7**
>
> Thanks for your valuable suggestions! Following are responses to your concerns.
>
> >**The importance of morphing between two natural images given only the start and end images is not clear. I suggest that the authors focus on the morphing of certain images instead of arbitrary natural images**
>
> We would like to note that **morphing between arbitrary image pairs** is not the main focus of our work. Instead, we assume that **the selected endpoint images should have certain perceptual connection for the interpolated images between them to be meaningful**, which is reflected from our sample selection strategy (as shown in sec 5.3 Appendix). Specifically,
> * For inter-class image pairs, we select benchmark images according to some LPIPS threshold.
> * For real-world images, the manually selected samples also have a certain perceptual connection.
> * In Fig.10 of the revised appendix, we provide a counterexample, showing that morphs between a pair of perceptually unrelated images can be meaningless.
> * Given the perceptual connection, we've also shown some examples where the morphs can be meaningful in novel content creation for different visual concepts, e.g., the morph between 'beetle' and 'car'.
> * Meanwhile, morphing between images belong to different categories has been studied for decades where morphing between human faces and animal faces are commonly applied image pairs. The second animation from [1] also shows an example of morphing between an ape and a bird. There are numerous morphing applications in entertainment industry, digital art, social media, etc.
>
> Thus we believe the perceptual connection between endpoint images (either decided by LPIPS threshold or users) can be a meaningful constraint to relate the endpoint images, which is a relaxation of traditional methods which use corresponding points to relate two images.
>
> >**[1] and [3-6] studies controllable image generation or image editing rather than image morphing.**
>
> [1] and [3-6] indeed deal with controllable generation or image editing.  We find that our image morphing method can be integrated with these methods to further strengthen their controllability. As shown in Fig.6 Appendix, given an image editing pipeline, our method can be used to allow users to control magnitude of editing strength, which is a desirable property in controllable generation and has been studied by previous works, e.g., [2].
>
> >**"Novel content creation", "data augmentation", and "model robustness analysis" mainly increase the diversity of datasets, for which image morphing may not be appropriate. Because there seems to be no need for directness.**
>
> * For novel content creation, directness can lead to higher level controllability, i.e., the user can control the morphing scale to generate the desired content based on selected end points while avoiding unexpected results. A direct morph also more aligns with human preference.
> * For data augmentation and model robustness analysis, given limited computational and labor resources, the expectation would be that diversity and relevance are both important. Lets consider a classification problem where majority of data stay away from the decision boundary while minority of data stay closed to the decision boundary. Creating a larger and more diverse dataset could generate more samples near the decision boundary to enhance the model robustness, but the cost is large in terms of computation and labelling. With our morphing method, we are able to generate more samples near the decision boundary with less cost because of directness.
>
> >**Regarding the adaptiveness of the rank, the paper empirically sets it using an empirically designed function. I would not say that it is adaptive because why to choose this formulation and how to set the hyper-parameters in the function are not clear.**
>
>
> Thanks for your suggestion, the rank selection equation is a heuristic we inferred from real-world data, where we find that the desired rank can vary from different image pairs. Empirically, we find that the proposed heuristic can provide an effective trade-off between directness (usually benefit from larger rank) and fidelity (usually benefit from smaller rank) for these samples.
>
>
> We acknowledge that this is heuristic. To avoid confusion, we are updating the term ‘adaptive strategy’ to ‘heuristic strategy’ in the main text. We will further investigate into the rank selection strategy and an interpretation of this problem in our future work.
>
> >**Probability measure on a manifold**
>
> Thank you for the correction, we have rectified this in the main text and response.
>
> Thank you once again for your valuable feedback. We hope the above responses have addressed your concerns.
>
> **References**
>
> [1] Wikipedia contributors. "Morphing." https://en.wikipedia.org/wiki/Morphing
>
> [2] Kwon, Mingi, Jaeseok Jeong, and Youngjung Uh. "Diffusion Models Already Have A Semantic Latent Space." The Eleventh International Conference on Learning Representations. 2022.

---

> ### Comment · Reviewer_UYJ7 · 2023-11-23
>
> I thank the authors for the effort in the rebuttal. After checking the revised paper and the appendix, I am convinced that the contribution and writing of the paper are near to the acceptance threshold of ICLR.

---

### Official Review · Reviewer_meGk · 2023-10-31

**Soundness:** 3 good
**Presentation:** 3 good
**Contribution:** 3 good
**Rating:** 6
**Confidence:** 4

**Summary:**

This paper presents an image morphing approach using pre-trained text-to-image diffusion models. The key idea is to interpolate text embeddings and latent states of two images following low-rank adaptation of model weights. The intuition is that the fine-tuned model exposes a more direct morphing path, yielding more perceptually convincing interpolations. Qualitative and quantitative experiments are performed to assess several key aspects of the proposed method. Overall, the method achieves superior morphing results compared to baselines and further supports interesting applications.

**Strengths:**

- The method makes use of a pre-trained text-to-image diffusion model for image morphing. The data prior captured by the diffusion model naturally constrains the morphing path on the image manifold, allowing perceptually convincing interpolations.

- The paper provides a helpful discussion on the tradeoffs in model adaptation. It discovers that LoRA creates an effective bottleneck to reduce overfitting, and further devises a simple strategy to calculate LoRA ranks based on sample diversity along the morphing path.

- The paper introduces an effective approach for perceptually uniform sampling. It is based on adaptive step size selection and yields smooth and visually pleasing morphing results.

- Several metrics are defined to evaluate different aspects of image morphing, namely smoothness, realism and directness. The method outperforms baselines in qualitative and quantitative experiments, and shows potential for several interesting applications.

**Weaknesses:**

- Despite the impressive results, there are some vague claims between the lines which need further clarification.

a) Equation 5 formulates image morphing as walking on the geodesic path in the 2-Wasserstein sense. However, the actual implementation amounts to linear interpolation of CLIP text embeddings, which in no way reflects the theoretical formulation.

b) For the interpolation of latent states, the paper cites an early work saying x0 and xT forms an optimal transport mapping. How does this justify the slerp interpolation between xT(0) and xT(1)?

- The method only compares to Wang and Golland (2023), another diffusion-based method for image morphing, yet there are numerous methods that are not based on diffusion (or even not learning-based). I am curious about how the method compares to, for example, frequency domain morphing methods, and what are their typical failure modes.

**Questions:**

- It would be helpful if the authors can comment on the robustness of their method relative to the hyper-parameters in model adaptation and sampling.

- Most of the image pairs used for experiments seem to have certain degree of spatial alignment. It would be helpful to know the rule of thumb for the selection of these image pairs.

---

> ### Author Response · Authors · 2023-11-17
> **Response to reviewer meGK regarding theoritical formulation, slerp, comparison, robustness and endpoint image selection.**
>
> Thanks for your concerns and instructive feedbacks. Following are some responses to your questions and concerns.
>
> >**W1: Equation 5 formulates image morphing as walking on the geodesic path in the 2-Wasserstein sense. However, the actual implementation amounts to linear interpolation of CLIP text embeddings, which in no way reflects the theoretical formulation.**
>
> Equation 5 is formulated following the previous work [1], which is used to reflect the desiderata of image morphing, i.e., the Wasserstein Barycenter problem leads to smoothness and directness of the transition between two images, while the constraint that the morphs should stay on a target data manifold leads to the realism of the morphs. However, it is extremely challenging to apply such formulation to high dimensional real world image data due to computational complexity as well as modeling of manifold; thus, previous work [1] restricts themselves to image data with simple background. We instead utilize alternative, tractable methods, without losing track of the desirable characteristics suggested by the theoretical problem. That is, based on the local linearity of the CLIP embedding space, we perform textual inversion and linear interpolation between inverted text embeddings and latent states which ensures realism and smoothness of interpolation. Our model adaptation strategy then enhances the directness of the morphing process while largely preserve the manifold geometry. Thus, although Equation 5 is not directly solved, we find that our work addresses the same criteria in a more tractable way.
>
> >**W2: The paper cites an early work saying x0 and xT forms an optimal transport mapping. How does this justify the slerp interpolation between xT(0) and xT(1)?**
>
> Thanks for your concern. Please refer to the general comment 2.
>
> >**W3: How the method compares to non-diffusion methods?**
>
> Please refer to the general comment 1.
>
> >**Q1: Robustness of method relative to the hyper-parameters in model adaptation and sampling.**
>
> Figure 4 of the main paper shows results of different LoRA ranks. In general, our method achieves reasonable performance across different LoRA ranks, while some ranks generate more visual appealing results than other ranks. If two samples are significantly different in visual appearance, model adaptation with large rank may lead to blurry intermediate samples.
>
> In terms of conditional guidance scales, we discover that, given fixed LPIPS difference for perceptually-uniform sampling, larger guidance scales are likely to generate longer sequences (less direct morphs). The reconstruction quality also decreases for large guidance scales, but overall the differences are not visually significant. We show an example of sequence generated from different guidance scale in Figure 14 of the revised appendix. Detailed results also report within tables in our appendix.
>
> >**Q2: The rule of thumb for the selection of these image pairs**
>
> Thanks for your question. Image pairs selected in our works are collected from three sources: (1) Example image pairs provided in Wang and Golland 2023; (2) Additional images that are manually collected from Internet; (3) Benchmark data for computer vision research. In terms of benchmark evaluation, we select the image pairs based on a simple protocol, i.e., for a given benchmark dataset, we randomly select image pairs whose similarity is under a certain LPIPS threshold, for details, please refer to sec 5.3 in the revised appendix. The intuition behind this process is that, the endpoint images should have certain perceptual connection for the interpolated images between them to be meaningful. In Fig.10 of the revised appendix, we provide a counterexample, showing that morphs between a pair of perceptually unrelated images can be meaningless and benchmark performances on these image pairs are not reliable.
>
> **References:**
>
> [1] Dror Simon and Aviad Aberdam. Barycenters of natural images constrained wasserstein barycenters for image morphing. In IEEE Conference on Computer Vision and Pattern Recognition (CVPR), pp. 7910–7919, 2020.

---

> > ### Comment · Reviewer_meGk · 2023-11-23
> >
> > Thanks for addressing my comments. Overall, the paper looks good to me, and I would like to keep my rating.

---

> ### Author Response · Authors · 2023-11-23
> **Thank you for your efforts**
>
> Thank you for your positive feedbacks and your valuable effort in reviewing our paper! Your comments helped clarify and improve our work.

---

### Official Review · Reviewer_zxkV · 2023-11-01

**Soundness:** 4 excellent
**Presentation:** 3 good
**Contribution:** 4 excellent
**Rating:** 6
**Confidence:** 4

**Summary:**

This paper proposes IMPUS, a diffusion-based image morphing method that can synthesize smooth and realistic transitions between two images. It employs a latent diffusion model with distinct conditional distributions for each image and interpolates in the locally linear and continuous text embedding space and Gaussian latent space. The method employs an adaptive bottleneck constraint based on a novel relative perceptual path diversity score to yield a direct path and suppresses ghosting artifacts in the interpolated images. Extensive experiments confirm IMPUS's effectiveness in image morphing and its potential for other image generation tasks.

**Strengths:**

The idea of applying diffusion models on image morphing task is interesting and promising;
The proposed evaluation criterion Smoothness,  Realism and Directness are reasonable and effective;
The presented techniques are effective and the overall method is able to generate high qulity results;
Ablation studies are thorough and abundant, which fully reveals the function and influence of each components of the model

**Weaknesses:**

In the comparison section, the proposed method is only compared with one previous method, which seems to be not comprehensive.  Also, there's only qualitative comparison, while quantitative criterion and user study would better convey the difference.
In the realism criteria and section3, the authors discussed about manifold and distribution several times, but gave no illustration or discussion in the experiment section.

**Questions:**

Why is data manifold and data distribution discuessed in methodologies but not covered in experiments? Would it be possible to visualize the data manifold or distribution with figures or give quantitative experiment resutls?
Why is the proposed method compared with only one existing diffusion based image interpolation method? Are there other similar works, can they be compared in fair experiment settings and criteria?

---

> ### Author Response · Authors · 2023-11-17
> **Response to reviewer zxkV regarding comparison study, quantitative analysis and data manifold**
>
> Thanks for the valuable suggestions and feedbacks. Following are some responses for your questions.
>
> >**W1: Only compared with one previous method, which seems to be not comprehensive**
>
> Please refer to general comment 1 for additional comparison with previous methods. Existing automatic morphing tools either create blurred and unrealistic intermediate images or cannot generalize to unseen data, while our work can generate smooth, direct and photorealistic morphs on real world images with large gaps.
>
> >**W2: There’s only qualitative comparison, while quantitative criterion and user study would better
> convey the difference**
>
> We thank the reviewer for the suggestion. We provide some quantitative comparison with Wang and Golland 2023 in Table 1 of the main paper (with their metrics and hyperparameters). We will defer the user study to future work.
>
>
> >**Q1: Data manifold and data distribution discussed in methodologies but not covered in experiments**
>
> We discuss the data manifold in the background section since many of the existing morphing works in the literature, which we mention in related work and background, are based upon ideas from manifold geometry and optimal transport to achieve smooth and realistic morphs. However, modeling manifold of high dimensional real world images are challenging, thus, these works restrict themselves to relatively simple image data. Some metrics reported in these works also not applicable for high dimensional image data, therefore, we propose some new metrics. The evaluation metrics proposed in paper are inspired by these works; thus, we mention these concepts to provide some intuition as to the selection of these metrics.
>
> In addition, the design of our method is heavily inspired by the hypothesis that high dimensional image data lie on low dimension manifolds. As pointed out by reviewer meGk, the data prior captured by the pre-trained diffusion model naturally constrains the morphing path on the image manifold and we apply low rank adaptation instead of standard finetuning to preserve the manifold learnt by the pre-trained diffusion model.
>
> To the best of our knowledge, it is challenging to obtain meaningful quantitative measurements related to the manifold or distribution of high dimensional images, so we defer this to future work.
>
> >**Q2: Why is the proposed method compared with only one existing diffusion based image interpolation method? Are there other similar works, can they be compared in fair experiment settings and criteria?**
>
> We note that the compared method is only existing diffusion-based work that is relavent to our targeted task. We provide additional comparison with some non-diffusion methods as illustrated in general comment 1.

---

### Official Review · Reviewer_NAVh · 2023-11-01

**Soundness:** 4 excellent
**Presentation:** 4 excellent
**Contribution:** 3 good
**Rating:** 6
**Confidence:** 4

**Summary:**

This paper proposes a new method to generate smooth and realistic image morphing results using diffusion models. The method starts with a baseline approach based on textural embedding interpolation and latent interpolation. To advocate direct and plausible interpolation results, authors fine-tune the diffusion model based on the image pairs with a bottleneck constraint implemented by LoRA to retrain the mode coverage along the morphing path. The rank is selected adaptively based on a metric that measures the density along the morphing path. To produce a smooth morphing, interpolation parameters are adaptively determined to ensure uniform perceptual transition. The proposed method demonstrates improved image morphing results compared to the prior work, especially when the semantics of the image pairs differ a lot.

**Strengths:**

- The method proposes several effective techniques to improve the prior diffusion-based baseline, yielding smooth, realistic and direct morphing results. The method works well on challenging cases as shown in Figure 1 and Figure 2 in the main text and Figure 2 in the Appendix, which are indeed impressive. Notably, these are achieved without auxiliary knowledge such as pose guidance as used in prior the method.

- The proposed techniques are simple yet effective. The LoRA finetuning effectively improves the plausibility of the interpolation results without mode collapse to training data. The perceptually uniform sampling based on the LPIPS difference is a quite useful technique and will benefit the following works in this area.

- The paper proposes novel metrics to measure the morphing quality in terms of directness, realism and smoothness. The quantitative comparisons are solid and sufficiently prove the effectiveness of the method.

**Weaknesses:**

- While the proposed method is effective, I think the contribution of the paper is relatively incremental. The framework is mostly built upon Wang and Golland 2023. Using LoRA to derive an input-aware diffusion model is relatively straightforward. And the proposed metrics are mostly based on prior work. Also, the LoRA rank adaption is heuristic and cannot be regarded as a major contribution.

- The physical meaning of the relative perceptual path diversity (rPPD) score is not clear. Why the score is a ratio that divides the
perceptual difference of the endpoint images? Moreover, the accumulated LPIPS difference cannot properly reflect the semantic difference of the input image pair. For example, the morphing for two bears that change a lot in location may be much easier than the morphing for two objects that differ significantly in semantics. To properly choose the LoRA rank, a much simpler metric that depends on CLIP score difference seems to suffice.

- Some of the proposed techniques only bring marginal improvement. As shown in Table 2, the best model is more likely not to be the one with adaptive rank. And in Figure 4, the difference between (n) and (o) is not significant.

- There is no quantitative study for the effectiveness of separate LoRA for unconditional score estimation. Also, there is no smoothness measure for perceptually uniform sampling.

**Questions:**

The rank adaption based on relative perceptual path diversity (rPPD) seems not effective enough. Is it possible to use much simpler heuristics based on CLIP score difference?

The techniques proposed in the paper do not make a significant difference when used independently, but the visual difference relative to the prior work (Figure 2) is impressive. It will be useful to show the qualitative results that show how the result progressively evolves by adding each technique one by one. Such analysis will let the readers understand which component plays the role at most.

---

> ### Author Response · Authors · 2023-11-17
> **Response to reviewer NAVh - Part 1/3**
>
> Thank you for the insightful comments. Following are responses for your questions.
>
> >**W1: Contribution of the paper is relatively incremental. The framework is mostly built upon Wang and Golland 2023**
>
> The major differences are as follows.
>
> Firstly, see general comment 4 for difference in task definition.
>
> Secondly, we are the first to propose that model adaptation is the key component for morphing. Despite its simplicity, this technique has been shown in our work to provide extraordinary improvements in both sample fidelity and directness. Meanwhile, unlike Wang and Golland 2023, the superior performance does not depend on careful selection for prompts, hyperparameters or random seeds. It also enables our method to morph inputs with pose inconsistency, while Wang and Golland 2023 relies on an auxiliary pose estimation model to handle this problem.
>
> Thirdly, the flexibility and robust performance shows the potential of our method to be adopted to a wide range of applications, e.g., in Sec. 5.3 of the main context, we show that our work can be adopted for model evaluation and video interpolation. We provide further examples in Fig.6 of the revised appendix showing that our work can be seamlessly incorporated with an existing state-of-the-art image editing \& translation pipeline to enable users for controlling the magnitude of change. Wang and Golland 2023 does not have such flexibility.
>
> Lastly, our sampling strategy is also different. We propose perceptually uniform sampling while Wang and Golland 2023 use standard uniform sampling. This has a significant impact on the quality of the results.
>
> A comparison with Wang and Golland 2023 is provided in Sec. 2 of the appendix. Given these comparisons, we believe that our work has significant improvement over Wang and Golland 2023 in terms of clearer task definition, more robust performance and more significant downstream benefits.
>
> >**W2: Using LoRA to derive an input-aware diffusion model is relatively straightforward. And the proposed metrics are mostly based on prior work. Also, the LoRA rank adaption is heuristic and cannot be regarded as a major contribution.**
>
> Our contributions can be summarized as follows: (1) we propose a robust open world automatic image morphing framework with diffusion models (see appendix for comparison with other existing methods); (2) we discover that low-rank adaptation contributes significantly to morphing task; (3) we present an algorithm for perceptually-uniform sampling that encourages the interpolated images to have visually smooth changes; (4) we propose three evaluation criteria, namely smoothness, realism, and directness, with associated metrics to measure the quality of the morphing sequence; and (5) we explore potential applications of morphing in broader domains such as video interpolation, model evaluation as well as data augmentation.
>
> Traditional image morphing is a labor intensive task as it requires involvement from human experts. Thus, creating a morphing sequence with good image quality is costly which restricts the applications of morphing in fewer domains. Some efforts for automatic image morphing have been performed; however, existing tools are only able to handle image pairs with small gaps. We are the first work which can robustly and automatically generate morphing sequences for image pairs with large gaps in semantics, pose, etc. Thus, we believe the technical contributions of our work are significant, as well as its ability to be used easily in applications.
>
> >**W3: The physical meaning of the relative perceptual path diversity (rPPD) score is not clear.**
>
> Please refer to general comment 3.

---

> ### Author Response · Authors · 2023-11-17
> **Response to reviewer NAVh - Part 2/3**
>
> >**W4: No quantitative study for separate LoRA for unconditional score estimation**
>
> Our main contribution is not applying separate LoRA for unconditional score estimation; thus, we do not perform comprehensive evaluation on that. During our experiments, we observed that using separate LoRA for unconditional score estimation performed more stable (compared with randomly discarded conditioning during finetuning) for image inversion, e.g., on CelebA (female-male), maximum end point reconstructed error in terms of LPIPS is **0.137** for separate LoRA unconditional score estimates, the value for randomly discarded conditioning during finetuning is **0.211**. A poor reconstruction may lead to inferior quality; thus, we select separate LoRA for unconditional score estimation. We show two examples of reconstructed images in Fig.13 of the revised appendix. To enhance the difference, we blend real as well as reconstructed images, i.e., $0.5 \times real + 0.5 \times recon$ in Fig. 13 (b). Other approaches could improve reconstruction quality as well, but applying separate LoRA for unconditional score estimates is simple yet effective enough for our task.
>
> >**W5: In Table 2, the best model is more likely not to be the one with adaptive rank. In Fig 4, the difference between (n) and (o) is not significant.**
>
> The adaptive rank equation is a heuristic inferred from real data. These images are quite different in terms of semantic, pose, etc., compared with the benchmark dataset. Most of benchmark datasets do not have such complexity, and are already partially aligned in pose and similar in semantics, or have already been seen by the pretrained diffusion model. Thus, issues such as mode collapse and blurred images may not be a large concern, which leads to a preference of higher rank. This observation is mentioned in the main paper.
>
> For Figure 4 (n)-(o) of the main paper, separate LoRA for unconditional score estimation leads to slightly more natural and balanced color in cheeks compared to randomly discard conditioning during finetuning. This phenomenon is related to the reconstruction quality. We show examples of reconstructed images from both methods in Fig.13 of the revised appendix.
>
> >**W6: No smoothness measure for perceptually uniform sampling**
>
> The total LPIPS ($LPIPS_T$) can serve as an indicator for the smoothness of perceptually uniform sampling. We provide some rationale for how the total LPIPS relates to (1) the rate of change between consecutive images and (2) the perceptual path length (PPL) below.
>
> Given two images $x^{(0)}$ and $x^{(1)}$, a diffusion model $x^{(\alpha_i)} = \mathcal{F}(\alpha_i, x^{(0)}, x^{(1)})$ generates an interpolated sample (between $x^{(0)}$ and $x^{(1)}$) based on the interpolation parameter $\alpha_i$. For a specified small delta LPIPS value $\Delta_{LPIPS}$ (which is almost imperceptible by human), perceptually uniform sampling ensures that $LPIPS(x^{(\alpha_i)}, x^{(\alpha_{i+1})})=\Delta_{LPIPS}$ and the morphing process requires $N$ interpolated samples to start from $x^{(0)}$ to $x^{(1)}$.
>
> We define this set of $N$ (non-uniform) interpolation scale as $A=\\{\alpha_1,...,\alpha_{N}\\}$, and a set of $N$ uniform interpolation scale as $B=\\{\frac{1}{N+1}, \frac{2}{N+1},...,\frac{N}{N+1}\\}$, then perceptually uniform sampling can be considered as a function $\mathcal{G}:B\to A$ such that $\mathcal{G}(\frac{i}{N+1})=\alpha_i$, and the generation of interpolated samples (given perceptually uniform sampling) can be written as $x^{(\alpha_i)} = \mathcal{F}(\mathcal{G}(\beta_i), x^{(0)}, x^{(1)})$ where $\beta_i \in B$. Thus, **rate of change** $\frac{\Delta_{LPIPS}}{\Delta_{\beta}}$ is always $(N+1)\cdot \Delta_{LPIPS}$ which is the **total LPIPS** along the morphing sequence. One noteworthy observation is that $N$ and $\Delta_{LPIPS}$ are not independent: as $N$ increase, $\Delta_{LPIPS}$ will decrease. The intuition behind is that for a fixed number of $N$ interpolated samples (from $x^{(0)}$ to $x^{(1)}$), models with larger total LPIPS will have a larger rate of change (in term of LPIPS) between two consecutive samples (less smooth given fixed number of interpolated points).
>
> Another metric for smoothness is **PPL**: $PPL_\epsilon = E_{\alpha \sim U(0,1)}[\frac{1}{\epsilon^2} \operatorname{LPIPS}(x^{(\alpha)}, x^{(\alpha+\varepsilon)})]$. Following notations from the previous paragraph, if we let $\varepsilon=\frac{1}{N+1}$, PPL for perceptually uniform sampling is $PPL_\epsilon = E_{\alpha \sim U(0,1)}[(N+1)^2 \cdot \Delta_{LPIPS}] = (N+1)^2 \cdot \Delta_{LPIPS} = \frac{(total LPIPS)^2}{\Delta_{LPIPS}}$. Standard PPL calculation requires $\varepsilon=\frac{1}{N+1} \ll 1$. In terms of perceptually-uniform sampling, $N$ should be defined in a manner such that $\Delta_{LPIPS}$ are almost imperceptible by human. Using total LPIPS to approximate PPL may not be precise due to numerical approximation, but the trend should be similar (larger total LPIPS indicates larger PPL).

---

> ### Author Response · Authors · 2023-11-17
> **Response to reviewer NAVh - Part 3/3**
>
> >**Q1: The accumulated LPIPS difference cannot properly reflect the semantic difference of the input image pair. CLIP score instead?**
>
> The accumulated LPIPS is not a measurement for semantic similarity, instead it measures the **sample diversity** of morphs which reflects the **perceptual directness** of a morphing result, which is the main criteria for selection of different ranks. The sample diversity is not solely decided by the gap in semantic information, it also depends on the gap in structural information, pose differences, overall style differences, etc., which are intractable to be directly measured by a metric. Thus, instead of measuring this information from the endpoints, we take a more flexible strategy by calculating the total perceptual change from a preliminary morphing result based on uniform sampling.
>
> To further clarify, we provide the cosine distance (1.0 - cosine\_similarity) in CLIP embedding space for samples in Fig.4 (a)-(l) of the revised main paper. For example, the pair of semantically similar samples in Fig.4 (a)-(f) (the flower images) with small cosine distance (0.15) have a more diverse morphing path than a pair of semantically dissimilar samples in Fig.4 (g)-(i) (the beetle car images) with large cosine distance (0.37). As can be seen in these cases, our proposed rPPD is more informative in selection of LoRA ranks than CLIP score. Please refer to general comment 3 for more details as well.
>
> >**Q2: Qualitative results that show how the result progressively evolves by adding each
> technique one by one**
>
> Thanks for the suggestion. In Fig.5 of the revised appendix, we provide the results of progressively adding proposed components in our work to the baseline of a pre-trained Stable Diffusion model, which uses user-provided prompts and discards the textual inversion and model adaptation: 1) Baseline, 2) Textual Inversion, 3) Textual Inversion + Vanilla Adaptation, 4) Textual Inversion + LoRA Adaptation (rank=4), 5) Textual Inversion + LoRA Adaptation (adaptive rank), 6) Textual Inversion + LoRA Adaptation (adaptive rank) + Unconditional Bias Correction.
>
> We can see that the baseline model fails to reconstruct one of the endpoint images and produces low fidelity samples. Textual inversion ensures a faithful reconstruction of the input endpoints. Model adaptation with LoRA can improve the perceptual directness of morphing path without suffering from the severe artifacts of vanilla model adaptation. An adaptively selected rank using the proposed rPPD metric shows improved directness than a commonly used rank, avoiding unnatural factor variations, e.g., using rank 4, the brightness of the left man's shirt first increases then decreases, while when adaptive rank is used, the brightness varies naturally. Results with the proposed unconditioned bias correction strategy shows finer details (such as hands and clothes). We've added discussion of this part in Sec. 3 of the appendix.

---

> > ### Comment · Reviewer_NAVh · 2023-12-04
> > **Response from Reviewer NAVh**
> >
> > Thanks for the response. The authors did a good job to solve my concerns. I would raise my rating and lean positive to the acceptance of the paper.

---

### Author Response · Authors · 2023-11-17
**General comments - Part 2/2**

>**3. The physical meaning of the *rPPD* score.**

**Fidelity** and **directness** are two important criteria for our morphing task, but there is a trade-off between them: morphing with larger rank adaptation is likely to be more direct but may suffer from blurriness due to overfitting, meanwhile an image pair with larger perceptual difference can suffer from this issue more easily. The design of rPPD score is base on the trade-off between fidelity (higher quality morphs) and directness (smaller total perceptual change). **Directness**: The **accumulated LPIPS**, which measures the directness of preliminary morph, is proportional to the LoRA rank for adaptation. **Fidelity**: A larger **perceptual difference between the endpoint images** can lead to higher risk in quality loss after larger rank adaptation, and is thus inversely proportional to the LoRA rank for adaptation. We use the ratio to control the trade-off between the two, and a larger rPPD indicates a larger LoRA rank can be applied to make the morphs more direct.

>**4. The difference between our work and Wang and Golland 2023 in *task definition*.**

The task definition of Wang and Golland 2023 focuses on **interpolation and novel image generation** but not **directness**, which is an important factor for image morphing. An indirect morph could generate a sequence of intermediate images irrelevant to the end point images as shown in Fig.2 of the revised Appendix, which also restricts downstream applications. **Directness of our method can lead to many downstream benefits**, e.g., for video interpolation, it facilitates temporal coherency and consistency (Fig.6 of the main paper); for interpolation of an image and it's edited version, it can align the image transition with the target editing direction allowing intuitive user control over the magnitude of change (Fig.6 of the revised appendix); for data augmentation, it limits the image variations to be related to the endpoint images, making the interpolated images to be relevant and useful.

**References:**

[1] Khrulkov, Valentin, et al. "Understanding ddpm latent codes through optimal transport." arXiv preprint arXiv:2202.07477 (2022).

[2] Chigozie Nri. Differentiable morphing. https://github.com/volotat/DiffMorph, 2022.

[3] András Jankovics. Automatic image morphing. https://github.com/jankovicsandras/autoimagemorph, 2020.

[4] David Dowd. Python image morpher (pim). https://github.com/ddowd97/Python-Image-Morpher, 2022.

[5] Sanghun Park, Kwanggyoon Seo, and Junyong Noh. Neural crossbreed: neural based image metamorphosis. ACM Trans. Graph., 39(6):224:1–224:15, 2020.

[6] Ferenc Huszár. Gaussian distributions are soap bubbles. https://www.inference.vc/high-dimensional-gaussian-distributions-are-soap-bubble/, 2017.

---

### Author Response · Authors · 2023-11-19
**General comments - Part 1/2**

We thank all the reviewers for the constructive comments. Particularly, **Reviewer NAVh** mentioned that *"The method works well on challenging cases as shown in Figure 1 and Figure 2 in the main text and Figure 3 in the Appendix, which are indeed impressive"* and *"The quantitative comparisons are solid and sufficiently prove the effectiveness of the method"*. **Reviewer zxkV** stated that *"The idea of applying diffusion models on image morphing task is interesting and promising."* **Reviewer meGk** acknowledged that *The method makes use of a pre-trained text-to-image diffusion model for image morphing. The data prior captured by the diffusion model naturally constrains the morphing path on the image manifold, allowing perceptually convincing interpolations*. **Reviewer UYJ7** noted that *Image morphing of nature images seems visually interesting to me*, further acknowledged our new task. We have revised the main context and the appendix to address the reviewers' concerns. The major revisions are marked in red.

We provide some general clarification for better understanding of both our task and our method.

>**1. Comparison with other morphing methods.**

We compare our morphing method with following open source automatic image morphing methods: DiffMorph [2] (Nri, 2022 in Appendix), Automatic Image Morphing [3] (Jankovics, 2020 in Appendix) (adapted from Python Image Morpher [4] (Dowd, 2022 in Appendix)) as well as Neural Crossbreed [5] (Park, et al., 2020 in Appendix) and show the examples in Fig.4 of the revised Appendix. Automatic Image Morphing is non-learning based, DiffMorph is optimization-based while Neural Crossbreed is GAN-based. Since Neural Crossbreed requires image pairs relevant to the GAN training data, we only compare with them on the image pair provided in their paper. According to Fig.4(a)-(c) of the revised appendix, morphs generated by our method are smoother and more realistic compared with DiffMorph and Automatic Image Morphing. In Fig.4(d) of the revised appendix, we show a failure case from Neural Crossbreed where the GAN model fails due to lack of training data. In summary, existing automatic morphing tools either create blurred & unrealistic intermediate images or cannot generalize to unseen data.

>**2. The paper cites [1] saying x0 and xT forms an optimal transport mapping. How does this justify the slerp interpolation between xT(0) and xT(1)?**

Thanks for the concern, we agree that this expression is ambiguous, we've rephrased this part and have added a further discussion in the appendix as follows

The finding in [1] shows that $x_0$ and $x_T$ forms an Monge optimal transport in the L2 sense, this suggests that the similarity in latent state $x_T$ will also cause certain pixel-wise similarities between the generated image $x_0$. As shown in Figure 4 of [1], such similarity in latent state indeed reflect the structural information in generated images. We provide similar results in Fig.8 of the revised appendix, where we fix the latent state $x_T$ for generations using a Stable Diffusion model with different text prompts, from where we can see significant alignment between the structure of generated images, which coincide with the finding of [1].

This finding indicates that latent state $x_T$ can reflect rich structural information about $x_0$, which makes it essential to correctly interpolate $x_T$ in order to smoothly morph two images. We use slerp instead of lerp to ensures that the interpolation can preserve the Euclidean norm of the interpolated latent state $x_T$ [6]. Empirically, we find slerp works well while lerp often produces unnatural intermediate samples since the Euclidean norm of interpolated latent states are not preserved in lerp. See Fig.9 of the revised appendix for examples.

---

### Author Response · Authors · 2023-11-20
**Nov 19 update**

As the discussion deadline (Nov 22, UTC-12) approaches, we kindly invite the reviewers to check out the revised version of the paper and the appendix, which provides additional experimental analysis and clarifications. Detailed responses have been provided to answer the reviewers' concerns and questions point by point. Please feel free to reach out if you have any further concerns or questions.

Best regards,

Authors

---

### Author Response · Authors · 2023-11-22
**[Gentle reminder] One day remains before the end of the discussion date**

Dear reviewers and AC,

We sincerely appreciate the valuable time you dedicated to reviewing our paper and providing constructive feedback. We have made every effort to revise the paper in accordance with your comments and have thoroughly answered your questions. We genuinely hope these revisions have effectively addressed your concerns. As **the discussion end date of November 22** approaches, please let us know if you have any further questions or concerns.

Best regards,

Authors

**Update:** We rephrase the term "adaptive rank" to "heuristic rank" in the main text to avoid confusion.

---

### Meta-Review · Area_Chair_Ldi1 · 2023-12-08

**Metareview:**

The paper introduces a diffusion-based approach aimed at automatic realistic and seamless image morphing from paired images. The inherent data captured by the diffusion model naturally confines the morphing trajectory on the image manifold, facilitating perceptually convincing interpolations. Additionally, it outlines a technique for achieving perceptually uniform sampling along the morphing path, incorporating an adaptive bottleneck constraint derived from a novel relative perceptual path diversity score. Notably, the method achieves a direct path and effectively mitigates ghosting artifacts in the interpolated images.

The results are very impressive, particularly in challenging scenarios where there is a significant difference in the semantics of the image pairs. The approach introduces several highly effective techniques to enhance the baseline, which relies on prior diffusion. This augmentation results in morphing outcomes that are not only smooth and realistic but also direct. The application of diffusion models to the image morphing task is both intriguing and promising. The data prior captured by the diffusion model naturally constrains the morphing path on the image manifold, leading to perceptually convincing interpolations. Furthermore, the utilization of perceptually uniform sampling, grounded in the LPIPS difference, proves to be a valuable technique. It is anticipated that this approach will prove beneficial for subsequent works in this area.

The majority of reviews raise the concern that there is only one previous method for comparison. In response, the rebuttal provides additional assessments of various automatic image morphing techniques. The supplementary results clearly demonstrate the strengths of the proposed method. There is also a major concern regarding the technical contributions, as the proposed method is based on Wang and Golland 2023. The rebuttal highlights the differences and contributions. There is a difference in the target tasks. Wang and Golland's work focuses on image interpolation, while this paper explores image morphing. As a means of accomplishing the directness goal for morphing, the paper introduces novel approaches, such as model adaptation for morphing and perceptually uniform sampling.

**Justification For Why Not Higher Score:**

The proposed method is built upon previous work. Although sufficiently novel modules are proposed and augmented along the way, the level of technical contributions does not meet the level of ICLR highlight papers.

**Justification For Why Not Lower Score:**

Results are excellent with realistic and smooth transitions between input image pairs, particularly for challenging cases. Furthermore, the method is automatic and robust.

---

### Decision · Program_Chairs · 2024-01-16

Accept (poster)